 Select

# Quantum eigenstates from classical Gibbs distributions

**Pieter W. Claeys[1*] and Anatoli Polkovnikov[2]**

**1** TCM Group, Cavendish Laboratory, University of Cambridge, Cambridge CB3 0HE, UK
**2** Department of Physics, Boston University, Boston, MA 02215, USA

⋆ pc652@cam.ac.uk

## Abstract

We discuss how the language of wave functions (state vectors) and associated non-commuting Hermitian operators naturally emerges from classical mechanics by applying the inverse Wigner-Weyl transform to the phase space probability distribution and observables. In this language, the Schrödinger equation follows from the Liouville equation, with $\hbar$ now a free parameter. Classical stationary distributions can be represented as sums over stationary states with discrete (quantized) energies, where these states directly correspond to quantum eigenstates. Interestingly, it is now classical mechanics which allows for apparent negative probabilities to occupy eigenstates, dual to the negative probabilities in Wigner's quasiprobability distribution. These negative probabilities are shown to disappear when allowing sufficient uncertainty in the classical distributions. We show that this correspondence is particularly pronounced for canonical Gibbs ensembles, where classical eigenstates satisfy an integral eigenvalue equation that reduces to the Schrödinger equation in a saddle-point approximation controlled by the inverse temperature. We illustrate this correspondence by showing that some paradigmatic examples such as tunneling, band structures, Berry phases, Landau levels, level statistics and quantum eigenstates in chaotic potentials can be reproduced to a surprising precision from a classical Gibbs ensemble, without any reference to quantum mechanics and with all parameters (including $\hbar$) on the order of unity.



# 1 Introduction

Quantum mechanics and classical mechanics differ not just in the physics they describe, but also in the mathematical ways they are generally expressed. Following Lagrange and Hamilton, classical mechanics is usually expressed in terms of time-dependent phase space variables, such as either coordinates and momenta or wave amplitudes and phases, and the dynamics is determined by Hamilton's equations of motion. Quantum mechanics in its non-relativistic formulation is generally expressed in the language of operators acting on states, where canonical variables are replaced by non-commuting operators. The time-dependence is then absorbed in either the states (Schrödinger representation) or the operators (Heisenberg representation). The non-commutativity of the operators leads to the fundamental quantum uncertainty relation set by Planck's constant $\hbar$. Classical uncertainties arise when considering e.g. an ensemble of particles, for which the equation of motion for the probability distribution is given by Liouville's equation. Similar ensembles of quantum particles are described by density matrices satisfying von Neuman's equation of motion.

It is possible to formally consider the limit $\hbar \to 0$, where the fundamental quantum uncertainties become small and classical mechanics can be recovered (see e.g. Refs. [1–4]). One way of observing this emergence is through the Wigner-Weyl phase space language [5–10]. This effectively rewrites the quantum equations of motion in terms of classical phase space

variables, which we will denote as $x$ and $p$ for concreteness[1], and von Neumann's equation for the Wigner (quasiprobability) function at $\hbar \to 0$ reduces to Liouville's equation for a classical probability distribution.

However, the reverse seems to be much less known: the classical Liouville equation can be rewritten in the language of state vectors (wave functions), and truncating third- and higher-order derivatives in the equations of motion leads to the Schrödinger equation. On the level of equations, the inverse Wigner transform of a probability distribution describing an ensemble of classical particles leads to a Hermitian *quasi*-density matrix. The Liouville equation now naturally returns von Neumann's equation of motion for this quasi-density matrix, with its eigenstates satisfying the time-dependent Schrödinger equation. This mapping and the subsequent emergence of the Schrödinger equation appears to have been (re)discovered at various occasions [11–18], where the original derivation seems to be first presented by Blokhintsev, as reviewed in Ref. [14]. One of the aims of the first part of the current paper is also to give a pedagogical motivation for the introduction of state vectors and Hermitian operators in quantum mechanics through a detailed overview of the emergent operator-state representation of classical mechanics.

However, and as should be expected, there are some crucial differences between this representation of classical mechanics and quantum mechanics. First, Planck's constant here appears as a scale that can be freely chosen and can be taken to be arbitrarily small. The proposed mapping to a quasi-density matrix and the operator representation is exact at all values of $\hbar$, but the resulting equations of motion only return the Schrödinger equation provided the initial classical distribution is sufficiently smooth on the scale of $\hbar$. In this sense, an initial condition corresponding to a single phase space point, i.e. fixed initial position and momentum, can not be properly described by the Schrödinger equation for generic Hamiltonians. Second, the weights arising in the classical formulation of the density matrix, which would correspond to probabilities in an actual density matrix, can be negative. Negative probabilities similarly appear in Wigner's quasiprobability function, and the negative probabilities here can be seen as dual to the ones in Wigner's function. As also discussed by Feynman, such negative (conditional) probabilities do not pose a problem as long as the probabilities of verifiable physical events remain positive [19]. These negative probabilities are in fact necessary if we want to avoid classical uncertainty relations, as also observed in Ref. [20]. Furthermore, in much the same way that the negative probabilities in the Wigner function disappear upon some averaging over phase space, we show that negative probabilities in classical mechanics disappear upon the introduction of sufficient uncertainty on the phase space distribution, in particular in the energy.

In the second part of the paper we present original results, which to the best of our knowledge did not appear anywhere before, and focus on the operator-state representation of canonical distributions. Remarkably, and this is perhaps the main finding of our paper, eigenstates of the canonical (classical) Gibbs ensemble are shown to satisfy an integral equation, which in turn reduces to the stationary Schrödinger equation in the saddle-point approximation controlled by the inverse temperature $\beta$. This result has the advantage that the accuracy of the approximation is set by both $\hbar$ and the temperature, where we show that a remarkably accurate correspondence between classical and quantum eigenstates can be obtained even for relatively large values of $\hbar$. This correspondence is shown to hold even in the presence of magnetic fields and for both symmetric (boson-like) and anti-symmetric (fermion-like) states. We illustrate these results with several paradigmatic examples, where we reproduce nearly-exact quantum wave functions and energy levels from the classical Gibbs ensemble even in the absence of particularly small parameters.

---

[1]To shorten notations we will use a two-dimensional phase space in most of this paper unless explicitly mentioned otherwise.

As an initial illustration, in Fig. 1 we present various classical eigenstates for the canonical Gibbs ensemble for a single-particle system in a two-dimensional double-well potential. These are visually indistinguishable from the eigenstates obtained by solving the Schrödinger equation, where we have chosen to present the two lowest-energy states and a single higher-excited state. The two lowest states correspond to the expected symmetric and antisymmetric combination of localized states, whereas the other state represents generic excited states.

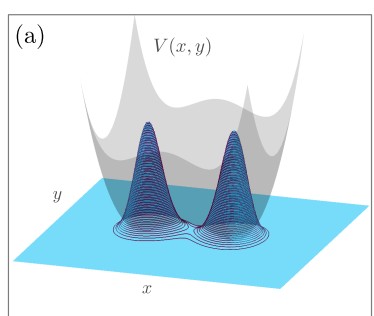 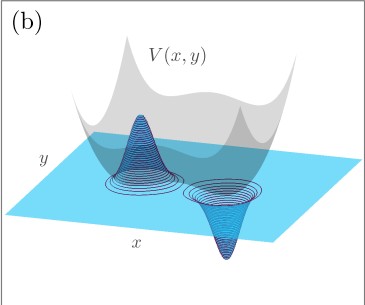 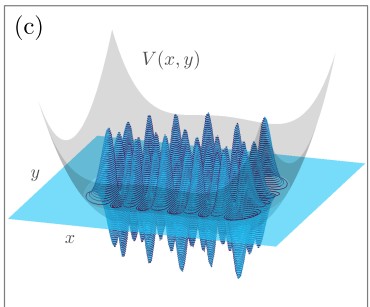

Figure 1: **Illustration comparing classical eigenstates obtained from the Gibbs distribution (blue surface plots) and quantum eigenstates obtained solving the stationary Schrödinger equation (black wire mesh) for a two-dimensional potential.** Two lowest-lying states (a) and (b) and a generic excited state (c) are shown for a two-dimensional potential (gray surface plot) corresponding to $V(x,y) = \frac{\nu}{4}(1-x^2)^2 + \frac{1}{2}m\omega^2 y^2$ with $m = \omega = \nu = 1$, inverse temperature $\beta = 0.1$, and $\epsilon = \hbar = 0.1$. See Section 4 for more details.

This paper is organized as follows. In Section 2, we provide a short recap of relevant results from the Wigner-Weyl formalism, after which an overview of the operator-state representation for classical mechanics is presented in Section 3, discussing the classical Schrödinger equation, the representation of classical observables as operators, and arguing for the necessity of negative probabilities. Then in the second half of the paper, we focus on the eigenstates of stationary canonical (Gibbs) distributions. Section 4 illustrates how the eigenvalue equation can be reduced to the stationary Schrödinger equation through a saddle-point approximation at high temperatures. Various examples of eigenstates and eigenvalues for the Gibbs ensemble are shown in Section 5. These include the harmonic oscillator, the quartic potential, and the double-well and periodic potential, where it is shown how tunneling states can be obtained, before concluding with some examples of eigenstates for two-dimensional potentials with and without magnetic fields. Equivalently, these two-dimensional systems can be viewed as representing interacting one-dimensional two-particle ensembles. Having obtained spectral properties for classical systems, Section 6 extends the Bohigas-Giannoni-Schmit and Berry-Tabor conjectures for the level statistics of quantum systems to classical systems. Section 7 is reserved for conclusions.

## 2 Wigner-Weyl formalism

For completeness, we first provide a short overview of the Wigner-Weyl formalism [6, 9, 10]. Readers familiar with this formalism can immediately skip to Sec. 3.

Given a quantum wave function $\psi(x, t)$, the Wigner function is defined as

$$W(x, p, t) = \int \frac{d\xi}{2\pi\hbar} \psi^*(x + \xi/2, t)\psi(x - \xi/2, t) \exp\left[\frac{i}{\hbar}p\xi\right]. \tag{1}$$

More generally, for a density matrix $\hat{\rho}(t)$,

$$W(x,p,t) = \int \frac{d\xi}{2\pi\hbar} \langle x+\xi/2|\hat{\rho}(t)|x-\xi/2\rangle \exp\left[\frac{i}{\hbar}p\xi\right] \tag{2}$$

$$= \int \frac{d\kappa}{2\pi\hbar} \langle p+\kappa/2|\hat{\rho}(t)|p-\kappa/2\rangle \exp\left[-\frac{i}{\hbar}x\kappa\right]. \tag{3}$$

This function has the important property that it returns the correct marginal distributions for the canonical variables

$$\langle x|\hat{\rho}(t)|x\rangle = \int dp\, W(x,p,t), \qquad \langle p|\hat{\rho}(t)|p\rangle = \int dx\, W(x,p,t). \tag{4}$$

This clearly suggests that $W$ could be interpreted as a joint probability distribution for $x$ and $p$. However, while this function is real and normalized, it is not necessarily positive. Rather, it belongs to a class of quasiprobability distributions.

The equation of motion for the Wigner function is highly similar to the classical Liouville equation, namely

$$\frac{\partial W}{\partial t} = \{H_{\mathrm{W}}, W\}_M \equiv \frac{2}{\hbar} H_W \sin\left(\frac{\hbar}{2}\Lambda\right) W, \tag{5}$$

where $H_{\mathrm{W}} \equiv H_{\mathrm{W}}(x,p)$ is the Weyl symbol of the quantum Hamiltonian $\hat{H}$, essentially a correctly-ordered classical Hamiltonian describing the particle[2], $\{\cdot,\cdot\}_M$ is the Moyal bracket, and $\Lambda$ is the symplectic operator, or loosely speaking simply the Poisson bracket operator:

$$\Lambda \equiv \frac{\overleftarrow{\partial}}{\partial x}\frac{\overrightarrow{\partial}}{\partial p} - \frac{\overleftarrow{\partial}}{\partial p}\frac{\overrightarrow{\partial}}{\partial x} \qquad \text{such that} \qquad A\Lambda B \equiv \frac{\partial A}{\partial x}\frac{\partial B}{\partial p} - \frac{\partial A}{\partial p}\frac{\partial B}{\partial x} \equiv \{A,B\}. \tag{6}$$

In the limit $\hbar \to 0$ the sin-function can be linearized, $\sin(\hbar\Lambda/2) \to \hbar\Lambda/2$, and von Neumann's equation (5) reduces to the Liouville equation [6,9,21]

$$\frac{\partial W}{\partial t} = \{H_{\mathrm{W}}, W\} + O(\hbar^2), \tag{7}$$

such that in this limit the function $W$ is conserved on continuous trajectories defined through the classical equations of motion

$$\frac{dx}{dt} = \{x, H_{\mathrm{W}}\} = \frac{\partial H_{\mathrm{W}}}{\partial p}, \qquad \frac{dp}{dt} = \{p, H_{\mathrm{W}}\} = -\frac{\partial H_{\mathrm{W}}}{\partial x}. \tag{8}$$

In this formulation of quantum mechanics, the key differences with classical mechanics are (i) the non-positivity of the Wigner function, not allowing a straightforward interpretation of $W$ as a probability distribution, and (ii) with the exception of harmonic systems, the impossibility of representing solutions of Eq. (5) through smooth characteristics $x(t)$ and $p(t)$ on which $W$ is conserved in time. In other words, while the Wigner function itself smoothly changes in time, this change can not be represented by smooth phase space trajectories of the particle, necessitating so-called quantum jumps.

## 3 Operator-State representation of the classical Liouville equation

Before presenting the full derivation, we will present some of the key results of the following sections. Starting from a classical probability distribution $P(x,p)$, a quasi-density matrix $\mathcal{W}$

---

[2]Which is obtained from Eq. (2) if we replace $\hat{\rho}$ with $\hat{H}$.

can be defined as

$$\mathcal{W}(x + \xi/2, x - \xi/2) = \int dp \, \exp\left[-i\frac{p\xi}{\epsilon}\right] P(x,p), \tag{9}$$

where we have introduced a dimensionful parameter $\epsilon$ that will play the role of $\hbar$. Switching to variables $(x_1, x_2) = (x + \xi/2, x - \xi/2)$, we can write

$$\mathcal{W}(x_1, x_2) = \sum_\alpha w_\alpha \psi_\alpha^*(x_1)\psi_\alpha(x_2), \tag{10}$$

which holds for any choice of parameter $\epsilon$, such that both the states and weights have an implicit dependence on $\epsilon$. The coefficients $w_\alpha$ are real and satisfy $\sum_\alpha w_\alpha = 1$ for normalized states $\psi_\alpha(x)$.

Classical averages over observables can be represented as Hermitian operators acting on the wave functions $\psi_\alpha(x)$, returning the coordinate representation of $x$ and $p$

$$x \to x, \qquad p \to -i\epsilon\partial_x, \tag{11}$$

where operator averages are calculated in the usual way. This operator representation always holds, irrespective of the choice of $\epsilon$ and without any approximation. Making the time-dependence explicit, in the limit $\epsilon \to 0$ (the precise scale determining this limit will be made clear in the derivation), the equation of motion for $\mathcal{W}$ given a Hamiltonian

$$H(x,p) = \frac{p^2}{2m} + V(x) \tag{12}$$

can be rewritten to show that all weights $w_\alpha$ are time-independent and the states $\psi_\alpha(x)$ satisfy the Schrödinger equation

$$i\epsilon\frac{\partial}{\partial t}\psi_\alpha(x,t) = \hat{H}\psi_\alpha(x,t) \qquad \text{with} \qquad \hat{H} = -\frac{\epsilon^2}{2m}\frac{\partial^2}{\partial x^2} + V(x). \tag{13}$$

Both in quantum and classical mechanics an important role is played by stationary, i.e. time-independent, distributions corresponding to stationary states. In classical mechanics such distributions are usually associated with statistical ensembles, with canonical and micro-canonical ensembles predominant in systems with a conserved number of particles. Applying the expansion (10) to a stationary distribution $\mathcal{W}(x_1, x_2)$ naturally leads to classical stationary states. A crucial result of this paper is that the stationary states corresponding to a canonical Gibbs ensemble at inverse temperature $\beta$ have a striking resemblance to quantum stationary states. As shown in Section 4, the exact eigenvalue equation for such canonical eigenstates can be written as

$$w_\alpha\psi_\alpha(x) = \frac{1}{Z_x}\int d\xi \, \exp\left[-\frac{m\xi^2}{2\beta\epsilon^2} - \beta V(x - \xi/2)\right]\psi_\alpha(x - \xi), \quad Z_x = \int dx \, \exp[-\beta V(x)], \tag{14}$$

which is similarly shown to return the Schrödinger equation when evaluating the integral using a saddle-point approximation controlled by $\beta$, with the eigenvalues returning the expected Boltzmann weights. Interestingly, this close correspondence between quantum and classical states remains even in the absence of small parameters (see Sec. 5). This analogy can similarly be extended to systems in the presence of a magnetic field, where the classical Gibbs distribution also returns the expected quantum Hamiltonian. We also note that this equation can be extended to multi-dimensional, multi-particle systems by adding coordinate/particle indices

to the $x$ and $\xi$ variables. In particular, it applies to systems of identical particles, returning symmetric and anti-symmetric classical eigenstates.

Of course, one immediate difference between quantum and classical mechanics in this language is that the parameter $\epsilon$ playing the role of $\hbar$ is arbitrary. There are, however, other key differences highlighting the complementarity of the quantum and classical formalisms. In particular, the $w_\alpha$ can be interpreted as quasiprobabilities, and it is now classical mechanics that can lead to negative probabilities of occupying states, defined as eigenfunctions of $\mathcal{W}(x_1, x_2)$. For the Gibbs ensemble we find that at high temperatures the weights $w_\alpha$ are all positive and coincide with the Boltzmann factors $w_\alpha \propto e^{-\beta E_\alpha}$, forming a broad distribution. In the opposite limit of vanishing temperatures $\beta \to \infty$, where the differences between quantum and classical states are most pronounced, the distribution of these weights is oscillatory and broad, with $w_\alpha \propto (-1)^\alpha e^{-\tilde\beta E_\alpha}$, with $\tilde\beta \propto 1/\beta$. The distribution is maximally narrow at some specific temperature $\beta^{-1}$ on the order of the ground state energy. It is precisely this negativity of probabilities that leads to the violation of the uncertainty principle in classical systems for small temperatures or, more generally, for narrow phase space probability distributions. Despite these subtleties, this "state language" of formulating classical mechanics is very useful as it provides both practical and conceptual tools for understanding the connections and differences between quantum stationary states and classical equilibrium distributions; quantum and classical chaos and integrability, entanglement and more. As one example, we show that the level spacing statistics of the eigenvalues $w_\alpha$ return either Wigner-Dyson or Poissonian statistics for chaotic and integrable Hamiltonians respectively. This connection also can help us to understand in what way some of the postulates of quantum mechanics are simply reformulations of classical results in the state language.

We also note that this approach is fundamentally different from the Koopman-von Neumann (KvN) formulation of classical mechanics [22, 23]. Within the KvN approach, a classical probability is written as the square of the absolute value of a wave function depending on both position and momentum. Classical observables are then represented by commuting operators, and time evolution of the KvN wave function is generated by a Liouvillian. In the approach outlined here, a classical probability distribution is first Fourier-transformed to $\mathcal{W}$, which can then be written as a density matrix with wave functions depending only on position. Classical observables are represented by generally non-commuting operators, and in the limit of sufficient uncertainty on the initial probability distribution the time evolution of the wave functions is generated by a Hamiltonian.

## 3.1 Classical Schrödinger equation.

We start this section by showing how the Liouville equation[3] for the classical probability distribution $P(x, p, t)$ [21]

$$\frac{\partial P}{\partial t} = -\{P, H\} = -\frac{\partial H}{\partial p}\frac{\partial P}{\partial x} + \frac{\partial H}{\partial x}\frac{\partial P}{\partial p}, \tag{15}$$

can be rewritten entirely in the coordinate space. This derivation essentially follows that of Refs. [14, 24], but because it is not widely known we will repeat it here for completeness. To simplify the notations, we consider a Hamiltonian describing a classical particle in a one-dimensional potential

$$H = \frac{p^2}{2m} + V(x), \tag{16}$$

where $m$ is the mass of the particle and $V(x)$ is the potential in which it moves. The assumption of a one-dimensional/single-particle system will be unimportant in the following, and

---

[3]Note the different sign compared with Hamilton's equations of motion for observables.

this derivation is readily generalized to more general Hamiltonians $H(x,p)$ that are arbitrary analytic functions of the phase space variables. Substituting the Hamiltonian (16) into the Liouville equation we find

$$\frac{\partial P}{\partial t} = -\frac{p}{m}\frac{\partial P}{\partial x} + V'(x)\frac{\partial P}{\partial p}. \tag{17}$$

It is convenient, by taking the Fourier transform with respect to momentum, to go from a representation of $P$ in terms of position and momentum to a representation in terms of a pair of position coordinates

$$\mathcal{W}(x+\xi/2, x-\xi/2, t) = \int dp \, \exp\left[-i\frac{p\xi}{\epsilon}\right] P(x,p,t). \tag{18}$$

In order for $\xi$ to have the dimension of position, we also introduce a dimensionful parameter $\epsilon$, which is kept arbitrary for the time being but will end up playing the role of $\hbar$. The reason for choosing $\mathcal{W}$ to be a function of $x+\xi/2$ and $x-\xi/2$ will be apparent shortly. This construction can be inverted as

$$P(x,p,t) = \int \frac{d\xi}{2\pi\epsilon} \, \exp\left[i\frac{p\xi}{\epsilon}\right] \mathcal{W}(x+\xi/2, x-\xi/2, t). \tag{19}$$

Note that if $\mathcal{W}(x_1, x_2)$ is replaced by the quantum density matrix $\rho(x_1, x_2)$ and $\epsilon$ by $\hbar$, $P(x,p)$ becomes the corresponding Wigner function [6] (see also Sec. 2). We can formally regard Eq. (18) as the inverse Wigner transform of the classical probability distribution with an arbitrarily chosen Planck's constant.

Taking the Fourier transform of Eq. (17) and using partial integration to evaluate the second term on the right hand side results in the following equation of motion for $\mathcal{W}$:

$$\frac{\partial}{\partial t}\mathcal{W}\left(x+\frac{\xi}{2}, x-\frac{\xi}{2}, t\right) = -\frac{i\epsilon}{m}\frac{\partial}{\partial x}\frac{\partial}{\partial \xi}\mathcal{W}\left(x+\frac{\xi}{2}, x-\frac{\xi}{2}, t\right)$$
$$+ \frac{i\xi}{\epsilon}V'(x)\mathcal{W}\left(x+\frac{\xi}{2}, x-\frac{\xi}{2}, t\right). \tag{20}$$

This equation can be made explicitly symmetric by switching to new variables $(x_1, x_2) = (x+\xi/2, x-\xi/2)$,

$$i\epsilon\frac{\partial}{\partial t}\mathcal{W}(x_1, x_2, t) = -\frac{\epsilon^2}{2m}\left[\frac{\partial^2}{\partial x_2^2} - \frac{\partial^2}{\partial x_1^2}\right]\mathcal{W}(x_1, x_2, t)$$
$$- (x_1 - x_2)V'\left(\frac{x_1+x_2}{2}\right)\mathcal{W}(x_1, x_2, t), \tag{21}$$

where we also multiplied Eq. (20) by $i\epsilon$. Eq. (21) is exact and holds for any value of $\epsilon$.

Now we can make a crucial simplification and take $\epsilon$ to be sufficiently small: in this case, we can see from Eq. (18) that in order for $\mathcal{W}$ to be nonzero, we also need to consider $\xi = (x_1 - x_2)$ sufficiently small compared to $\epsilon$. Namely, if $P(x, p, t)$ does not significantly change when varying $p$ over a scale on the order of $\epsilon/\xi$, the integral over the exponential will average out to zero, and the only non-zero contributions to $\mathcal{W}$ will be for $\xi$ similarly small. We can make an approximation and set $\xi V'(x) \approx V(x+\xi/2) - V(x-\xi/2) = V(x_1) - V(x_2)$, such that the equation of motion reduces to

$$i\epsilon\frac{\partial}{\partial t}\mathcal{W}(x_1, x_2, t) \approx \left[-\frac{\epsilon^2}{2m}\frac{\partial^2}{\partial x_2^2} + V(x_2)\right]\mathcal{W}(x_1, x_2, t)$$
$$- \left[-\frac{\epsilon^2}{2m}\frac{\partial^2}{\partial x_1^2} + V(x_1)\right]\mathcal{W}(x_1, x_2, t). \tag{22}$$

Similar to regular quantum mechanics, Eq. (22) can be simplified through an eigenvalue decomposition of $\mathcal{W}(x_1, x_2, t)$, interpreting $\mathcal{W}$ as an operator with $(x_1, x_2)$ as indices. Note that this operator is Hermitian: taking its transpose, i.e. exchanging $x_1$ and $x_2$, corresponds to taking $\xi \to -\xi$ in Eq. (18), which is in turn equivalent to taking the complex conjugate. As such, this operator is guaranteed to be diagonalizable, with real eigenvalues. Then

$$\mathcal{W}(x_1, x_2, t) = \sum_\alpha w_\alpha \Psi_\alpha^*(x_1, t) \Psi_\alpha(x_2, t), \tag{23}$$

where the functions $\Psi_\alpha(x, t)$ are the analogues of the time-dependent wave functions in the Schrödinger representation and $w_\alpha$ are weights/eigenvalues. While the latter can in principle dependent on time, in the limit where Eq. (22) holds they are time-independent, while the orthonormal wave functions satisfy the Schrödinger equation

$$i\epsilon \frac{\partial}{\partial t} \Psi_\alpha(x, t) = \left[ -\frac{\epsilon^2}{2m} \frac{\partial^2}{\partial x^2} + V(x) \right] \Psi_\alpha(x, t). \tag{24}$$

## 3.2 Representation of observables through Hermitian operators.

The representation of the probability density as a (quasi-)density matrix immediately leads to the representation of observables as Hermitian operators acting on states. This mapping does not depend on any approximations, as we will now explore. For convenience, the time dependence of all distributions, states, and operators is also made implicit in this section. First of all, we can choose the eigenstates to be normalized as

$$\int dx\, \Psi_\alpha^*(x) \Psi_\beta(x) = \delta_{\alpha\beta}. \tag{25}$$

From the normalization of the classical probability distribution, we then find that

$$1 = \int dx \int dp\, P(x, p) = \int dx\, \mathcal{W}(x, x) = \sum_\alpha w_\alpha. \tag{26}$$

More generally, expressing the classical probability distribution $P(x, p)$ through the function $\mathcal{W}$, it is straightforward to find that

$$\langle x \rangle \equiv \int dx \int dp\, P(x, p)\, x = \sum_\alpha w_\alpha \int dx\, |\Psi_\alpha(x)|^2\, x, \tag{27}$$

$$\langle p \rangle \equiv \int dx \int dp\, P(x, p)\, p = \sum_\alpha w_\alpha \int dx\, \Psi_\alpha^*(x) \left[ -i\epsilon \frac{\partial}{\partial x} \Psi_\alpha(x) \right]. \tag{28}$$

From the first expression for $x$, one can see that $\mathcal{W}(x, x) = \sum_\alpha w_\alpha |\Psi_\alpha(x)|^2$ plays the role of the coordinate probability distribution. In fact, this correspondence holds for general observables that only depend on position, where for any function $O(x)$ one has

$$\langle O(x) \rangle \equiv \int dx \int dp\, P(x, p) O(x) = \int dx\, \mathcal{W}(x, x) O(x). \tag{29}$$

From Eq. (28) we see that the momentum is represented by the partial derivative

$$p \to \hat{p} = -i\epsilon \partial_x, \tag{30}$$

such that the operators $\hat{x}$ (represented by multiplication by $x$) and $\hat{p}$ satisfy the usual commutation relation

$$[\hat{x}, \hat{p}] = i\epsilon. \tag{31}$$

It is easy to check that this representation survives if we consider an expectation value of an arbitrary function $O(p)$. Note that the definition of $\mathcal{W}$ in Eq. (18) implied choosing the coordinate representation of wave functions. Alternatively, we could have obtained the momentum representation

$$\widetilde{\mathcal{W}}(p + \kappa/2, p - \kappa/2) = \int dx \, \exp\left[i\frac{x\kappa}{\epsilon}\right] P(x, p, t), \tag{32}$$

where the momentum representation of operators would give $p \to p$ and $x \to i\epsilon\partial_p$.

One can similarly analyze more complicated observables involving products of $x$ and $p$. For example,

$$
\begin{aligned}
\langle xp \rangle &\equiv \int dx \int dp \, P(x, p) x p \\
&= \frac{i\epsilon}{2} \sum_\alpha w_\alpha \int dx \, x \left( \frac{\partial \Psi_\alpha^*(x)}{\partial x} \Psi_\alpha(x) - \Psi_\alpha^*(x) \frac{\partial \Psi_\alpha(x)}{\partial x} \right) \\
&= \sum_\alpha w_\alpha \int dx \, \Psi_\alpha^*(x) \left( -x i\epsilon \frac{\partial}{\partial x} - \frac{i\epsilon}{2} \right) \Psi_\alpha(x),
\end{aligned}
\tag{33}
$$

where we obtained the last equation by integrating by parts. We see that, as perhaps expected,

$$xp \to \hat{x}\hat{p} - \frac{i\epsilon}{2} = \frac{\hat{x}\hat{p} + \hat{p}\hat{x}}{2}. \tag{34}$$

This equation immediately extends to functions of the form $pO(x)$, where

$$pO(x) \to \frac{\hat{p}\hat{O}(\hat{x}) + \hat{O}(\hat{x})\hat{p}}{2}. \tag{35}$$

It is straightforward to check that more general functions of the form $p^n O(x)$ correspond to so-called symmetrically-ordered operators (see e.g. Refs. [9,25–27]), which can be defined by the recursion relation

$$p^n O(x) \to \hat{\Omega}_n(\hat{x}, \hat{p}) = \frac{1}{2}\left[\hat{p}\,\hat{\Omega}_{n-1}(\hat{x}, \hat{p}) + \hat{\Omega}_{n-1}(\hat{x}, \hat{p})\,\hat{p}\right], \quad \text{where} \quad \hat{\Omega}_0(\hat{x}, \hat{p}) = \hat{O}(\hat{x}). \tag{36}$$

All symmetrically-ordered operators are explicitly Hermitian, which immediately follows from the recursion relation above, combined with the fact that $\hat{x}$ and $\hat{p}$ are Hermitian. As an important example, for a time-independent Hamiltonian the energy of a system is conserved and given by

$$\int dx \int dp \, P(x, p) H(x, p) = \sum_\alpha w_\alpha \int dx \, \Psi_\alpha^*(x) \left[ -\frac{\epsilon^2}{2m} \frac{\partial^2}{\partial x^2} + V(x) \right] \Psi_\alpha(x), \tag{37}$$

which is independent of the limit $\epsilon \to 0$ necessary to obtain the Schrödinger equation.

### 3.3 Negative probabilities for classical systems. The uncertainty principle.

Much of the previous section exactly reproduced the language of quantum mechanics entirely within a classical formalism. The only approximation we made was in the equation of motion determining the dynamics, setting

$$(x_1 - x_2)V'\left(\frac{x_1 + x_2}{2}\right) \approx V(x_1) - V(x_2), \tag{38}$$

when acting on $\mathcal{W}(x_1, x_2)$, as required to obtain Eq. (22) from the exact equation (21). This approximation is justified if the distribution $P(x, p)$ is sufficiently smooth on the scale set by $\epsilon$ such that $\mathcal{W}(x_1, x_2, t)$ is only non-zero when $|x_1 - x_2| = |\xi|$ is small enough. Note that for harmonic potentials there are no approximations involved and Eq. (22) is exact for any choice of $\epsilon$ since then $(x_1 - x_2)V'((x_1 + x_2)/2) = V(x_1) - V(x_2)$.

There seems to be an apparent contradiction with Heisenberg's uncertainty relation $\delta x \delta p \geq \epsilon/2$. In quantum mechanics this uncertainty relation is a direct consequence of the commutation relation (31). However, this commutation relation also holds classically, and even more, it holds for any choice of $\epsilon$. Yet, the initial distribution $P(x, p)$ can be chosen to be arbitrarily narrow and violate the uncertainty relation. To resolve this apparent paradox, it's necessary to conclude that $\mathcal{W}$ cannot be an exact density matrix: for a general distribution the weights $w_\alpha$ entering Eq. (23), playing the role of probabilities, will not all be positive. This allows us to interpret these weights as quasiprobabilities in the same way the non-positive quantum Wigner function $W(x, p, t)$ is interpreted as a quasiprobability in phase space, since the weights are real and sum to unity, $\sum_\alpha w_\alpha = 1$. We thus arrive at the interesting conclusion that, in the operator-state representation, it is now classical mechanics that leads to apparent negative probabilities. If we choose $\epsilon$ larger than $\hbar$, such that real quantum effects are neglected, but still small enough such that Eq. (21) holds, we can effectively realize density matrices with negative (quasi-)probabilities, which could lead to phenomena not possible in ordinary quantum mechanics.

As an immediate corollary, only classical distributions which satisfy the uncertainty relation $\delta x \delta p \geq \epsilon/2$ can have all non-negative weights $w_\alpha$. This can be exemplified from a Gaussian phase space distribution

$$P(x, p) = \frac{1}{2\pi\sigma_x\sigma_p} \exp\left(-\frac{x^2}{2\sigma_x^2} - \frac{p^2}{2\sigma_p^2}\right), \tag{39}$$

where the qualitative character of the eigenvalues will depend crucially on $\sigma_x \sigma_p$. First consider the distribution saturating this bound,

$$P(x, p) = \frac{1}{\pi\epsilon} \exp\left(-\frac{x^2}{2\sigma_x^2} - \frac{p^2}{2\sigma_p^2}\right), \qquad \sigma_x\sigma_p = \frac{\epsilon}{2}, \tag{40}$$

where $\mathcal{W}(x_1, x_2)$ can be easily obtained and shown to factorize as

$$\mathcal{W}(x_1, x_2) = \psi^*(x_1)\psi(x_2), \qquad \psi(x) = \frac{1}{(2\pi\epsilon^2)^{1/4}} \exp\left(-\frac{x^2}{4\sigma_x^2}\right). \tag{41}$$

The quasi-density matrix has a single non-zero eigenvalue and the corresponding eigenstate is the ground state of a harmonic oscillator with frequency $\omega = \epsilon/(2m\sigma_x^2) = (2\sigma_p^2)/(m\epsilon) = \sigma_p/(m\sigma_x)$.

Now consider the case where $\sigma_x\sigma_p > \epsilon/2$. The Wigner function for the Gibbs ensemble of a harmonic oscillator is exactly given by a Gaussian, which we can invert to show that in this case the quasi-density matrix following from Eq. (39) is given by (as shown in Appendix C.2)

$$\mathcal{W}(x_1, x_2) = \frac{1}{\mathcal{Z}} \sum_{n \geq 0} \exp[-n\beta_q\epsilon\omega]\psi_n^*(x_1)\psi_n(x_2), \qquad \mathcal{Z} = \frac{1}{1 - \exp(-\beta_q\epsilon\omega)}, \tag{42}$$

corresponding to the Gibbs distribution of a harmonic oscillator with frequency $\omega$ at inverse temperature[4] $\beta_q$; $\psi_n(x)$ is the $n$-th quantum eigenstate of this oscillator. The values of $\omega$ and

---

[4]We use $T_q$, $\beta_q$ to denote the temperature and its inverse of a quantum system to avoid confusion with $T$, $\beta$, which we use later for the classical Gibbs ensemble.

$\beta_q$ follow from the uncertainties as

$$m\omega = \frac{\sigma_p}{\sigma_x}, \qquad \sigma_x\sigma_p = \frac{\epsilon}{2}\coth\left(\frac{\epsilon\omega\beta_q}{2}\right). \tag{43}$$

Since $\sigma_x\sigma_p > \epsilon/2$, the second equation has a real and positive solution for $\beta_q$, such that all weights $w_n = \exp[-n\beta_q\epsilon\omega]/\mathcal{Z}$ are strictly positive and have a clear interpretation as probabilities.

The final case we can consider is for $\sigma_x\sigma_p < \epsilon/2$, i.e. when the distribution $P(x,p)$ *does not* satisfy the uncertainty relation. This can be obtained by shifting $\beta_q$ in Eq. (43) to $\beta_q = \tilde{\beta}_q + i\pi/(\epsilon\omega)$, such that

$$m\omega = \frac{\sigma_p}{\sigma_x}, \qquad \sigma_x\sigma_p = \frac{\epsilon}{2}\tanh\left(\frac{\epsilon\omega\tilde{\beta}_q}{2}\right), \tag{44}$$

where we have used $\coth(x + i\pi/2) = \tanh(x)$. Given a Gaussian with $\sigma_x\sigma_p < \epsilon/2$, these equations now have a real and positive solution for $\tilde{\beta}_q$. Introducing the same shift of $\beta_q$ in Eq. (42), we conclude that the eigenstates of the quasi-density matrix violating the uncertainty relation remain those of a harmonic oscillator with frequency $\omega$, but the weights will now become oscillatory and (see also Appendix C.2)

$$\mathcal{W}(x_1, x_2) = \frac{1}{\mathcal{Z}}\sum_{n\geq 0}(-1)^n \exp[-n\tilde{\beta}_q\epsilon\omega]\psi_n^*(x_1)\psi_n(x_2), \qquad \mathcal{Z} = \frac{1}{1 + \exp(-\tilde{\beta}_q\epsilon\omega)}, \tag{45}$$

since $\exp[-n\beta_q\epsilon\omega] = \exp[-n\tilde{\beta}_q\epsilon\omega]\exp[-in\pi] = (-1)^n\exp[-n\tilde{\beta}_q\epsilon\omega]$. Negative probabilities arise the moment the uncertainty relation is violated. Clearly, a narrow distribution $P(x,p)$ with $\sigma_x\sigma_p \ll \epsilon/2$ will correspond to a small $\tilde{\beta}_q$ and result in a very broad oscillatory distribution of the weights $w_n \propto (-1)^n\exp[-n\tilde{\beta}_q\epsilon\omega]$.

As an interesting observation, we note that a partition function of the form $\mathcal{Z} = 1/(1 - \exp(-\beta_q\epsilon\omega))$ naturally arises in the description of free bosons, whereas that of a free fermion is given by $(1 + \exp(-\tilde{\beta}_q\epsilon\omega))$. For the harmonic oscillator the eigenstate index $n$ can be interpreted as an occupation number, leading to an average energy of $\epsilon\omega(\langle n \rangle + 1/2)$, and it can easily be checked that

$$\langle n \rangle_{\sigma_x\sigma_p > \epsilon/2} = \frac{1}{\exp(\beta_q\epsilon\omega) - 1}, \qquad \langle n \rangle_{\sigma_x\sigma_p < \epsilon/2} = -\frac{1}{\exp(\tilde{\beta}_q\epsilon\omega) + 1}, \tag{46}$$

returning the Bose-Einstein and (minus the) Fermi-Dirac distributions respectively. While the bosonic Bose-Einstein distribution for oscillators with $\sigma_x\sigma_p > \epsilon/2$ is not unexpected, the analogy between free fermions and the quantum oscillator at a complex temperature, in turn equivalent to a Gaussian classical probability distribution with $\sigma_x\sigma_p < \epsilon/2$, is rather intriguing. At the moment we are not sure if this is a simple coincidence or if there is a deeper underlying reason.

## 4 Canonical distributions

Both in quantum and classical mechanics a special role is played by stationary states/ stationary probability distributions. In quantum mechanics these states are defined as eigenstates of the Hamiltonian. While all possible stationary probability distributions are not classified in classical systems, an important class of such distributions is those corresponding to equilibrium

statistical ensembles. For single-particle systems the two most common ensembles are given by the canonical (fixed temperature) and microcanonical (fixed energy) distribution. As we will see the classical-quantum correspondence is most pronounced for canonical ensembles, such that we will focus on these in the following. Results for the microcanonical ensemble are presented in Appendix B.

For the Hamiltonian given by Eq. (16) the canonical, or Gibbs or Maxwell-Boltzmann, probability distribution is given by

$$P(x, p) = \frac{1}{Z} \exp\left[-\beta\left(\frac{p^2}{2m} + V(x)\right)\right], \tag{47}$$

in which the normalization constant or partition function is given by

$$Z = \int dx \int dp \exp\left[-\beta\left(\frac{p^2}{2m} + V(x)\right)\right] = \sqrt{\frac{2\pi m}{\beta}} \int dx \exp\left[-\beta V(x)\right]. \tag{48}$$

It straightforward to compute the function $\mathcal{W}(x + \xi/2, x - \xi/2)$ for this distribution by explicitly taking the Fourier transform according to Eq. (9) as

$$\mathcal{W}(x + \xi/2, x - \xi/2) = \frac{1}{Z_x} \exp\left[-\frac{m\xi^2}{2\beta\epsilon^2} - \beta V(x)\right], \quad Z_x = \int dx \exp\left[-\beta V(x)\right]. \tag{49}$$

Since this is a stationary state by construction, $\mathcal{W}(x_1, x_2)$ should approximately commute with the Hamiltonian provided the assumptions of Section 3.1 hold, such that its eigenstates should aproximate the eigenstates of the quantum Hamiltonian. In the next Section we will show explicit examples of stationary eigenstates for particular potentials obtained in this way.

We can advance analytically by deriving the equation for the stationary eigenstates $\psi_n(x)$ and quasi-probabilities $w_n$. Writing out the eigenvalue equation and changing the integration variable $x_2$ to $\xi = x_1 - x_2$, we find the following *exact* integral equation:

$$w_n \psi_n(x) = \frac{1}{Z_x} \int d\xi \exp\left[-\frac{m\xi^2}{2\beta\epsilon^2} - \beta V(x - \xi/2)\right] \psi_n(x - \xi). \tag{50}$$

## 4.1 Saddle point derivation of the stationary Schrödinger equation.

As in the WKB method, it is convenient to define a complex action $S_n(x)$ as $\psi_n(x) = \exp[iS_n(x)/\epsilon]$. The above integral equation now reads

$$w_n \exp\left[\frac{i}{\epsilon} S_n(x)\right] = \frac{1}{Z_x} \int d\xi \exp\left[-\frac{m\xi^2}{2\beta\epsilon^2} - \beta V(x - \xi/2) + \frac{i}{\epsilon} S_n(x - \xi)\right]. \tag{51}$$

This equation can be simplified in the limit of small $\epsilon$, leading to essentially the same analysis as in Sec. 3.1. It is perhaps more interesting to note that the inverse temperature $\beta$ can serve as an alternative different saddle point parameter, since in the limit $\beta \to 0$ the first prefactor in the exponential diverges. Note that $\epsilon$ and $\beta$ do not enter the integral equation through a fixed combination, so the expansions in $\epsilon$ and $\beta$ do not coincide, distinguishing the proposed approach from quasi-classical methods. This difference will become apparent in the next section, where we derive the leading inverse-temperature corrections to the eigenvalue equation.

Performing the Taylor expansion of the integrand in $\xi$ up to the second order and integrating over $\xi$, which is equivalent to the saddle point approximation, we find

$$w_n = \frac{1}{Z_x} \sqrt{\frac{2\pi\beta\epsilon^2}{m}} \exp\left[-\beta\left(\frac{S_n'(x)^2}{2m} + V(x) - \frac{i\epsilon}{2m} S_n''(x)\right) + O(\beta^2)\right]. \tag{52}$$

Since the left-hand side of this equation is $x$-independent we must have

$$\frac{S_n'(x)^2}{2m} + V(x) - \frac{i\epsilon}{2m}S_n''(x) = \text{constant} = E_n, \tag{53}$$

where we have labelled the constant $E_n$. As is well known from the WKB analysis [28], this equation is equivalent to the stationary Schrödinger equation for the wave function with eigenvalue $E_n$,

$$-\frac{\epsilon^2}{2m}\frac{d^2\psi_n(x)}{dx^2} + V(x)\psi_n(x) = E_n\psi_n(x). \tag{54}$$

We also immediately see that the quasi-probabilities are, up to a prefactor, simply the Boltzmann weights of the discrete energies $E_n$

$$w_n = \frac{e^{-\beta E_n}}{\mathcal{Z}}, \qquad \mathcal{Z} = \sum_n e^{-\beta E_n} = \sqrt{\frac{m}{2\pi\beta\epsilon^2}}Z_x = \frac{1}{2\pi\epsilon}Z. \tag{55}$$

There is a clear connection between the classical canonical distribution and quantum stationary states. If we take $\epsilon = \hbar$ and analyze the eigenstates of $\mathcal{W}$, here the Fourier transform of the Gibbs distribution, the correct "quantum" stationary states are recovered. As we will show in Section 5, these include stationary states in a non-linear potential, tunneling states and random states in chaotic two-dimensional systems. Because the saddle point approximation is justified by the smallness of $\beta$ and not $\epsilon = \hbar$, the accuracy of classical eigenstates is independent from the accuracy of the WKB approximation, and as we will demonstrate one can very accurately recover both the ground and excited states. Moreover, as we numerically observe for smooth potentials, the difference between quantum and classical eigenstates remains surprisingly small even if $\beta$ is on the order of one and only few states are effectively populated. We also note that within the saddle point approximation all probabilities $w_n$ are strictly positive – a small enough $\beta$ such that the saddle point approximation holds implies a smooth enough phase space distribution such that all necessary uncertainty relations hold.

## 4.2 Inverse temperature expansion of the classical Gibbs Hamiltonian.

While the saddle-point derivation in the previous section is relatively straightforward and remarkably accurate, it remains important to obtain corrections determining the differences between quantum and classical eigenstates away from the limit $\beta \to 0$. The saddle-point approach is less convenient for finding finite $\beta$ corrections, where it is more convenient to develop a formal approach based on the operator expansion. This can also serve to highlight the difference between the expansion in $\beta$ and semiclassical approaches expanding in $\epsilon$ or $\hbar$.

To do so, it is convenient to rewrite Eq. (50) as

$$w_n\psi_n(x) = \frac{\epsilon}{\sqrt{2\pi}Z}\int d\chi\, \exp\left[-\frac{\chi^2}{2} - \beta V\left(x - \frac{\epsilon\chi}{2}\sqrt{\frac{\beta}{m}}\right)\right]\exp\left[-\epsilon\chi\sqrt{\frac{\beta}{m}}\frac{d}{dx}\right]\psi_n(x), \tag{56}$$

where we changed the integration variable $\xi \to \chi\epsilon\sqrt{\beta/m}$ to make the expansion in $\beta$ more transparent. We also represented $\psi_n(x - \xi) = \exp[-\xi d_x]\psi_n(x)$. We can formally define the effective classical Gibbs Hamiltonian as

$$\hat{H}_{\text{Gibbs}} = -\frac{1}{\beta}\log\left(\int\frac{d\chi}{\sqrt{2\pi}}\exp\left[-\frac{\chi^2}{2} - \beta V\left(x - \frac{\epsilon\chi}{2}\sqrt{\frac{\beta}{m}}\right)\right]\exp\left[-\epsilon\chi\sqrt{\frac{\beta}{m}}\frac{d}{dx}\right]\right)$$
$$= -\frac{1}{\beta}\log\left(\overline{\exp\left[-\beta V\left(x - \frac{\epsilon\chi}{2}\sqrt{\frac{\beta}{m}}\right)\right]\exp\left[-\epsilon\chi\sqrt{\frac{\beta}{m}}\frac{d}{dx}\right]}\right), \tag{57}$$

where the overline represents Gaussian averaging with respect to $\chi$ over the normal distribution with zero mean and unit variance. The eigenstates of this Gibbs Hamiltonian are the exact eigenstates of $\mathcal{W}$. This representation again makes clear that the expansion in $\epsilon$ and the expansion in $\beta$ necessarily differ, since $\beta$ enters as both the prefactor in front of the potential and as a scaling factor for the integration variable $\chi$.

Expanding the exponent to the leading order in $\beta$ we find

$$\hat{H}_{\text{Gibbs}} = -\frac{1}{\beta} \log\left(1 - \beta V(x) + \overline{\xi^2}\frac{\beta \epsilon^2}{2m}\frac{d^2}{dx^2} + O(\beta^2)\right) = \frac{\hat{p}^2}{2m} + V(x) + O(\beta), \qquad (58)$$

which is exactly the result of the saddle-point analysis. Evaluation of the next order corrections is tedious but straightforward, where we only quote the final result[5]:

$$\begin{aligned}
\hat{H}_{\text{Gibbs}} = {} & \frac{\hat{p}^2}{2m} + V(x) - \frac{\beta \epsilon^2}{8m}V''(x) \\
& + \frac{\beta^2 \epsilon^2}{24m}\left(\frac{1}{4m}\left[\hat{p}^2 V''(x) + 2\hat{p}V''(x)\hat{p} + V''(x)\hat{p}^2\right] + V'(x)^2\right) + O(\beta^3).
\end{aligned} \qquad (59)$$

This expression again highlights how the finite-temperature expansion in $\beta$ is not a semiclassical expansion in $\epsilon$: while both return the same zeroth-order Hamiltonian, the first two orders of correction in $\beta$ are clearly of the same order in $\epsilon$. It can be checked that this correction is trivial for the harmonic oscillator, where the linear correction represents an overall energy shift and the quadratic correction is proportional to the Hamiltonian. This multiplicative correction can in turn be absorbed by redefining the "quantum temperature" $\beta_q$ in agreement with the earlier results (see e.g. Eq. (43)). If the potential contains both harmonic and unharmonic parts one can, e.g., rescale $\beta_q$ in order to keep the harmonic part independent of the classical temperature. This procedure could result in nontrivial renormalization group-type flows of the Gibbs Hamiltonian and $\beta_q$ with $\beta$.

## 4.3 Canonical ensemble in the presense of a vector potential.

We now extend the previous discussion to a more general class of classical Hamiltonians, where a particle is coupled to an electromagnetic vector potential. Namely, let us now consider the following classical Hamiltonian[6]:

$$H = \frac{1}{2m}(p - qA(x))^2 + V(x), \qquad (60)$$

where $A(x)$ is the vector potential and $q$ the charge of the particle. Taking the Fourier transform of the corresponding Gibbs distribution, we find the following expression generalizing Eq. (49):

$$\mathcal{W}(x + \xi/2, x - \xi/2) = \frac{1}{Z_x}\exp\left[-\frac{m\xi^2}{2\beta \epsilon^2} - i\frac{\xi}{\epsilon}qA(x) - \beta V(x)\right], \qquad (61)$$

which leads to the following eigenvalue equation (cf. Eq. (50)):

$$w_n \psi_n(x) = \frac{1}{Z_x}\int d\xi \exp\left[-\frac{m\xi^2}{2\beta \epsilon^2} - i\frac{\xi}{\epsilon}qA(x - \xi/2) - \beta V(x - \xi/2)\right]\psi_n(x - \xi). \qquad (62)$$

Repeating, for example, the same analysis as in Eq. (56) up to leading order in $\beta$, we recover the correct quantum Hamiltonian of a particle in the presence of a vector potential:

$$\hat{H}_{\text{Gibbs}} = \frac{1}{2m}(\hat{p} - qA(x))^2 + V(x) + O(\beta^2). \qquad (63)$$

---

[5]We thank A. Dymarsky for helping us with this derivation.

[6]For simplicity, the derivation focuses on a one-dimensional system, but holds for systems with arbitrary dimension.

We see that in the high-temperature limit the eigenstates of the classical Gibbs distribution again return the correct eigenstates of the quantum Hamiltonian. As such, many quantum phenomena including the existence of a nontrivial Berry phase or Aharonov-Bohm-type interference of stationary eigenstates outside a magnetic solenoid are encoded in the classical Gibbs distributon. In Section 5.5 we will illustrate this statement with specific examples.

# 5 Examples of canonical stationary states

In this section we will analyze eigenstates of the canonical distribution in several characteristic single-particle systems with increasing complexity, and compare these with the corresponding quantum mechanical states. In particular, we will analyze a harmonic oscillator, a quartic potential, a double-well and periodic potential, and finally a two-dimensional non-linear potential with and without magnetic field.

## 5.1 Harmonic oscillator.

We will start from the harmonic oscillator, where all the eigenstates can be found analytically. The potential energy is then $V(x) = \frac{1}{2}m\omega^2 x^2$ such that the Boltzmann's distribution reads

$$P(x,p) = \frac{1}{Z}\exp\left[-\frac{\beta m\omega^2 x^2}{2} - \frac{\beta p^2}{2m}\right]. \tag{64}$$

This is precisely the Gaussian distribution we analyzed earlier (see Eq. (39)) with $\sigma_x = 1/\sqrt{\beta m\omega^2}$ and $\sigma_p = \sqrt{m/\beta}$ such that $\sigma_p/\sigma_x = m\omega$ and $\sigma_x\sigma_p = 1/(\beta\omega)$. From Eq. (43) we see that this distribution maps to the equilibrium canonical ensemble,

$$\mathcal{W}(x_1, x_2) = \sum_n w_n \psi_n(x_1)\psi_n(x_2), \tag{65}$$

where $\psi_n(x)$ are the eigenstates of the quantum harmonic oscillator with the same parameters and $\hbar \to \epsilon$, defined in terms of Hermite polynomials $H_n(x)$,

$$\psi_n(x) = \frac{1}{\left(\pi\xi_0^2\right)^{1/4}}\frac{1}{\sqrt{2^n n!}}H_n(x/\xi_0)\exp\left[-\frac{x^2}{2\xi_0^2}\right] \qquad \text{with} \qquad \xi_0^2 = \frac{\epsilon}{m\omega}, \tag{66}$$

and for $T = 1/\beta \geq \epsilon\omega/2$

$$w_n = \frac{1}{\mathcal{Z}}\exp[-\beta_q\omega n], \quad \beta_q = \frac{2}{\epsilon\omega}\operatorname{arctanh}\frac{\beta\omega\epsilon}{2}, \tag{67}$$

whereas for $T < \epsilon\omega/2$

$$w_n = \frac{(-1)^n}{\mathcal{Z}}\exp[-\tilde{\beta}_q\omega n], \quad \tilde{\beta}_q = \frac{2}{\epsilon\omega}\operatorname{arccoth}\frac{\beta\omega\epsilon}{2}. \tag{68}$$

As expected, for $T \gg \epsilon\omega/2$ we have $\beta_q \approx \beta$ and in the opposite limit $T \ll \epsilon\omega/2$ we have $\tilde{\beta}_q \approx 4/(\beta\epsilon^2\omega^2)$.

We see that for the harmonic potential the stationary eigenstates $\psi_n(x)$ coincide at any temperature $\beta$ with the eigenstates of a quantum harmonic oscillator if we set $\hbar \to \epsilon$. This result also follows from the fact that the canonical distribution is stationary by construction and Eq. (21) reduces to the commutator with the Hamiltonian with $\epsilon = \hbar$ without any approximation, such that $\mathcal{W}$ necessarily commutes with the Hamiltonian and they share a common

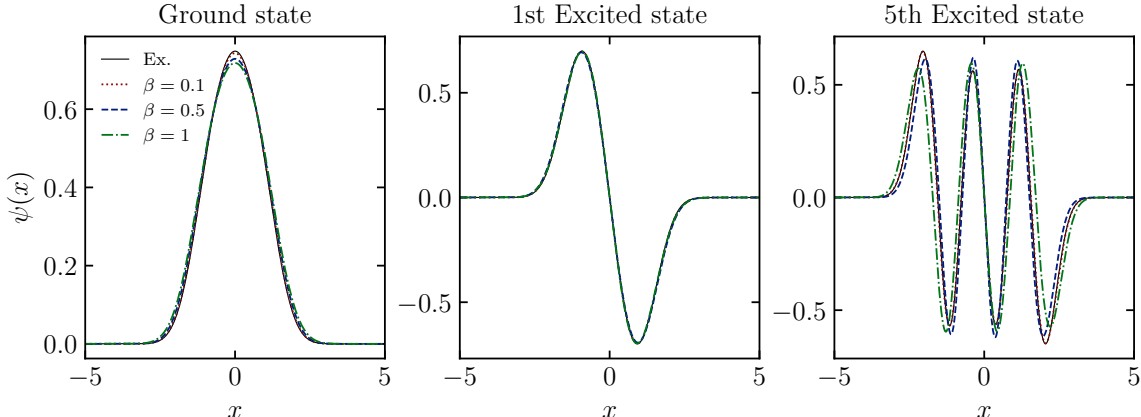

Figure 2: **Comparison of the classical eigenstates obtained from the Gibbs distribution and quantum eigenstates obtained solving the stationary Schrödinger equation for the quartic potential** $V(x) = vx^4/4$. States are shown at various $\beta = 0.1, 0.5, 1$. Other parameters: $m = 1, \epsilon = \hbar = 1, v = 1$.

eigenbasis. In this sense there is a precise correspondence between the classical Gibbs ensemble and the quantum Gibbs distribution for $T \geq \epsilon\omega/2$ if we identify the quantum inverse temperature with $\beta_q$ according to the relations above. Interestingly, for $T < \epsilon\omega/2$ we still have the exact correspondence between the classical and quantum probability distributions but the quantum weights $w_n$ are no longer positive, with an additional oscillatory dependence on top of the exponential decay.

## 5.2   Quartic potential.

Now we move to the slightly more complicated case of a particle of mass $m$ in a nonlinear quartic potential $V(x) = \frac{1}{4}vx^4$. The exact quantum ground state energy for this model can be found numerically as

$$E_{\text{gs}} \approx 0.421 \frac{\epsilon^{4/3} v^{1/3}}{m^{2/3}}. \tag{69}$$

This energy also provides a characteristic quantum energy scale for the system. We will now compare some eigenstates obtained from the classical Boltzmann's distribution with the quantum eigenstates at different values of $\beta$. For concreteness we will fix all the parameters $\epsilon, v, m$ to be unity.

In Fig. 2 we show examples of several wave functions describing the ground state, the first excited state, and the fifth excited state of the particle in the quartic potential. All plots illustrate a comparison between the exact quantum wave functions obtained by numerically solving the Schrödinger equation (full black lines) and the classical eigenstates obtained by diagonalizing $\mathcal{W}(x_1, x_2)$ corresponding to the Gibbs distribution at three different values of $\beta = 0.1,\ 0.5,\ 1$. In both the quantum and classical case we use standard diagonalization routines for symmetric matrices, taking identical spatial discretization steps which we choose sufficiently small such that we reproduce accurate continuous results. We checked that in all analyzed cases the diagonalization of the quantum Hamiltonian $\hat{H}$ and of the matrix $\mathcal{W}(x_1, x_2)$ takes similar amounts of time.

While at $\beta = 1$ there are clear visible differences between the classical and quantum states, the agreement between them is still strikingly good given that there is no single small parameter in the problem. The "high-temperature" classical eigenstates at $\beta = 0.1$ are visually indistinguishable from the quantum eigenstates. In Table 1 we list the energies of the classical

eigenstates, computed as usually as the expectation values of the Hamiltonian,

$$E_n = \langle \psi_n | \hat{H} | \psi_n \rangle = \int dx \, \psi_n^*(x) \left[ -\frac{\epsilon^2}{2m} \frac{d^2}{dx^2} + V(x) \right] \psi_n(x). \tag{70}$$

The second column gives the (numerically) exact quantum-mechanical spectrum and the three following columns describe the energy spectrum following from the eigenstates of the classical Gibbs eigenstates computed at three different temperatures. As with the wave functions, the last column corresponding to $\beta = 0.1$ gives nearly exact results with about 0.01% accuracy. The lower temperature spectrum has a larger discrepancy with the quantum spectrum but is still pretty accurate. It is remarkable that for $\beta = 1$ even the tenth eigenstate, with energy about 15 times larger than the classical temperature and with a tiny occupation, is still reproduced reasonably well.

Table 1: **Comparisons of energies of the first 10 eigenstates corresponding to a particle in a quartic potential.** The first column corresponds to the exact quantum energies. The next three columns are the energies of the classical eigenstates obtained from the Gibbs distribution at three different temperatures. The parameters are the same as in Fig. 2.

| State # | Quantum | Gibbs $\beta = 1$ | Gibbs $\beta = 0.5$ | Gibbs $\beta = 0.1$ |
|---|---|---|---|---|
| 1 | 0.420805 | 0.429898 | 0.423806 | 0.420976 |
| 2 | 1.5079 | 1.50845 | 1.50986 | 1.50814 |
| 3 | 2.95879 | 3.18835 | 2.95619 | 2.95889 |
| 4 | 4.62121 | 5.01404 | 4.62432 | 4.62122 |
| 5 | 6.45348 | 6.22582 | 6.48496 | 6.4534 |
| 6 | 8.42841 | 8.29998 | 8.5441 | 8.42822 |
| 7 | 10.5278 | 10.5521 | 10.898 | 10.5275 |
| 8 | 12.7382 | 13.4236 | 14.1554 | 12.7378 |
| 9 | 15.0496 | 15.113 | 15.2664 | 15.0491 |
| 10 | 17.4538 | 17.2797 | 15.9425 | 17.4531 |

In the left panel of Fig. 3 we show the distribution of quasi-probabilities $w_n$ for the first twenty classical eigenstates corresponding to the four different temperatures of the Gibbs ensemble. We emphasize that these probabilities, together with the corresponding eigenstates $\psi_n(x)$, *exactly* represent the classical Gibbs ensemble. At inverse temperature $\beta = 0.1$ the distribution of quasi-probabilities almost exactly matches the quantum Gibbs distribution at the same temperature, as illustrated in the right panel of Fig. 3. While qualitatively the similarities with the quantum ensemble extend all the way to $\beta = 1$, one can clearly observe the emergence of negative weights with increasing $\beta$. From this plot it is clear that it is possible to generate classical probability distributions dominated by the ground state and yet still have very good agreement with the corresponding quantum ground state. In other words, despite the absence of any intrinsic quantization, classical Gibbs ensembles have excellent discrete representations where a small number of eigenstates can be occupied. As the temperature is further lowered and the classical distribution becomes too narrow, such that it is no longer possible to satisfy the uncertainty principle, the discrete representation becomes broad again, with many states occupied and weights $w_n$ becoming strongly oscillatory (see the data for $\beta = 5$ in Fig. 3). This behavior of the quasi-probabilities is qualitatively very similar to that of the Gaussian phase space distribution discussed in previous section and the microcanonical distribution discussed in Appendix B.

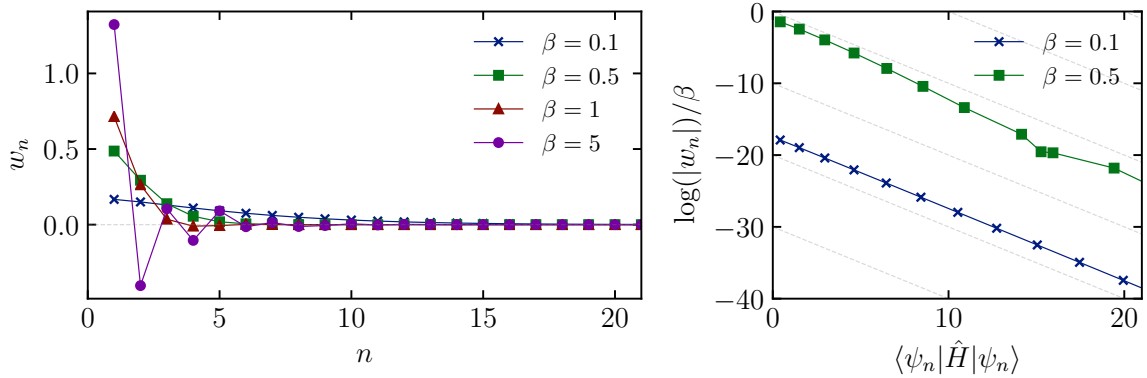

Figure 3: **Distribution of quasiprobabilities $w_n$ for the classical Gibbs ensemble for the quartic potential at four different temperatures.** The left plot shows the eigenvalues of $\mathcal{W}(x_1, x_2)$ at four values of $\beta$, where negative probabilities arise at larger $\beta$. The right plot shows $\log(|w_n|)/\beta$ as function of $E_n = \langle \psi_n | \hat{H} | \psi_n \rangle$ for the two smaller values of $\beta$. The parameters are the same as in Fig. 2.

In order to further quantify the behavior of the weights/eigenvalues/quasi-probabilities, we consider

$$S_\alpha = \frac{1}{1-\alpha} \log \left( \sum_n w_n^\alpha \right). \tag{71}$$

For positive eigenvalues, this corresponds to the Rényi entropy, which is bounded from below by zero and $S_\alpha = 0$ only for a factorizable distribution with a single nonzero $w_n = 1$. When allowing for negative $w_n$, this lower bound and the interpretation as entropy vanishes, but it is still instructive to consider how $S_\alpha$ changes as $\beta$ is varied. In order to avoid negative arguments for the logarithm, we consider $\alpha$ integer and even. $S_2$ can be analytically obtained as

$$S_2 = -\log \mathrm{Tr}\left[\mathcal{W}^2\right] = -\log \int dx_1 \int dx_2 \, \mathcal{W}(x_1, x_2) \mathcal{W}(x_2, x_1)$$

$$= -\log 2\pi\epsilon \int dx \int dp \, P(x, p)^2 = -\log \left[ \sqrt{\frac{\pi \beta \epsilon^2}{m}} \frac{\int dx \, \mathrm{e}^{-2\beta V(x)}}{\left( \int dx \, \mathrm{e}^{-\beta V(x)} \right)^2} \right]. \tag{72}$$

The partition function integrals can be explicitly evaluated for the harmonic oscillator to return $S_2 = -\log(\beta \epsilon \omega / 2)$ and for the quartic potential to return

$$S_2 = -\log \left[ \sqrt{\frac{\pi \epsilon^2}{m}} \frac{\beta^{3/4} (2\nu)^{1/4}}{4 \, \Gamma(5/4)} \right], \qquad \Gamma(5/4) = 0.9064\ldots, \tag{73}$$

such that $S_2$ is positive for $\beta^{-1} \gtrsim 0.485 \frac{\epsilon^{4/3} \nu^{1/3}}{m^{2/3}}$, close to the ground-state energy of Eq. (69).

For sufficiently small $\beta$ the saddle-point approximation holds, such that all weights are positive and $S_2 > S_4 > S_6$. Further increasing $\beta$, these entropies equal zero around (but not exactly at, see inset) the same value of $\beta \approx 2.06 \, m^{2/3} / (\epsilon^{4/3} \nu^{1/3})$. Near this point, the distribution is close to factorizable, with a dominant eigenvalue $w_0 \approx 0.990$ and second and third largest eigenvalues $0.108$ and $-0.081$. Further increasing $\beta$, all $S_\alpha$ become negative with inverted ordering $S_2 < S_4 < S_6$, indicating the expected presence of negative eigenvalues. The same behavior would be observed for the harmonic oscillator, where $S_2$ becomes negative at $\beta^{-1} = \epsilon \omega / 2$, where the distribution is exactly factorizable and the uncertainty relation

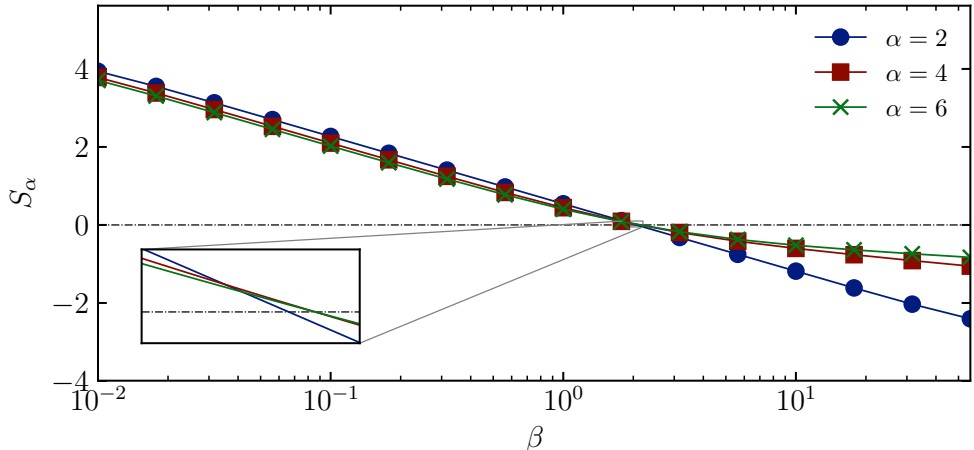

Figure 4: **Entropies $S_\alpha$ for increasing $\beta$ for the quartic potential.** Parameters are the same as in Fig. 2, such that $S_2 = 0$ at $\beta \approx 2.06$.

is satisfied. More generally, given a fixed classical distribution it should also be possible to choose $\epsilon$ in such a way that the discrete representation is optimized, minimizing the number of non-negligible weights, and it would be interesting to identify such a choice that returns the actual Planck's constant as $\epsilon = \hbar$.

## 5.3 Double-well potential and periodic potential.

Next we consider a particle of mass $m$ in a somewhat more complicated double-well potential,

$$V(x) = \frac{\nu}{4}(x^2 - 1)^2. \tag{74}$$

As before, we will choose $m = 1$ for the numerical analysis, although we now consider a smaller value of $\epsilon = \hbar = 0.1$ as for larger values of $\epsilon$, e.g. $\epsilon = 1$, the potential is too weak to support tunneling states [7]. In Fig. 5 we show the classical and quantum wave functions corresponding to the lowest symmetric and antisymmetric tunneling states (top) and to the second pair of symmetric and anti-symmetric tunneling states (bottom) at various values of $\beta$. Already for $\beta = 1$, the classical and the quantum states are visually indistinguishable.

To quantify the accuracy of the agreement between the quantum and classical tunneling states in Fig. 6, we plot the relative error in the tunneling gap between both pairs of tunneling states as a function of $\beta$. This relative error is defined as

$$\frac{\Delta_{\text{QM}} - \Delta_\beta}{\Delta_{\text{QM}}} \equiv 1 - \frac{E_\beta(n+1) - E_\beta(n)}{E_{\text{QM}}(n+1) - E_{\text{QM}}(n)}, \tag{75}$$

where the eigenstate index $n = 1$ corresponds to the two lowest tunneling states and $n = 3$ describes the second pair of states. The index $\beta$ implies that the corresponding energies are obtained from the classical Gibbs distribution at the inverse temperature $\beta$, where we calculate classical energies as the expectation value of the Hamiltonian $\hat{H}$ w.r.t. the eigenstates of $\mathcal{W}$, and the index QM corresponds to the numerically calculated quantum eigenstates of the Hamiltonian. The mistake clearly increases with $\beta$ and scales as $\beta^2$, but even at $\beta = 1$ the relative error is still less than 0.5%, which is surprisingly accurate given that the tunneling splitting

---

[7]By introducing a dimensionless length in units of $\ell = (\epsilon^2/m\nu)^{1/6}$ and a dimensionless energy in units of $\Delta = \epsilon^{4/3}\nu^{1/3}/m^{2/3}$ one can reduce the number of independent parameters in the quantum problem to one: $\sigma = a/\ell$. In the classical Gibbs ensemble there is an extra dimensionless parameter set by the temperature: $\beta\Delta$.

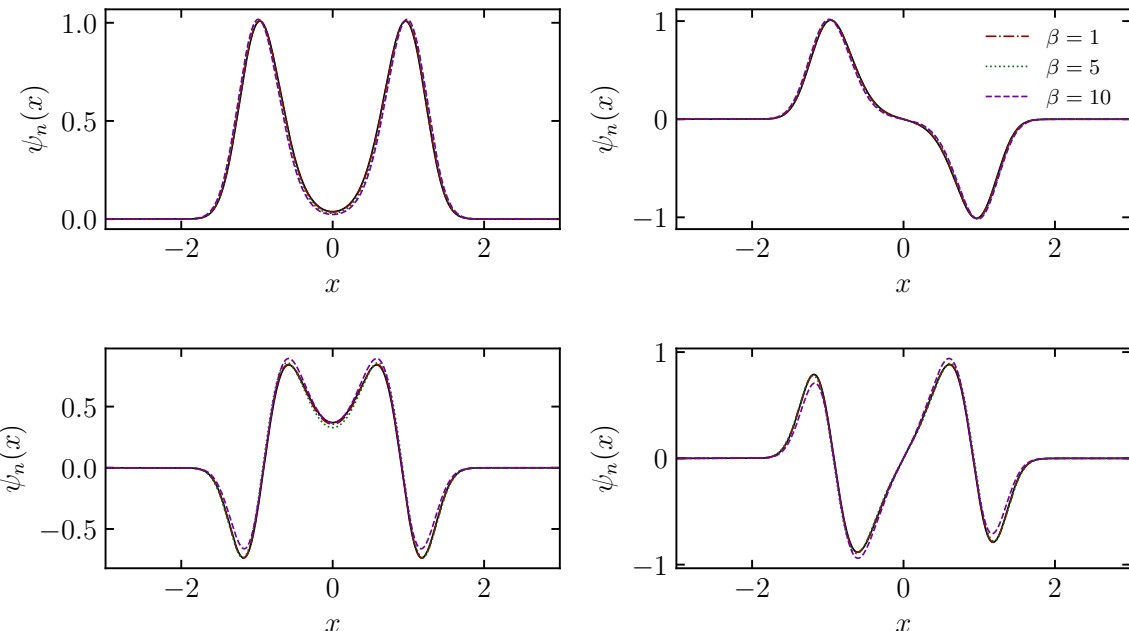

**Figure 5: Comparison of the classical and quantum lowest energy tunneling states (top) and the next two tunneling states (bottom) for the double well potential** $V(x) = v(x^2 - 1)^2/4$. The inverse temperature is $\beta = 1, 5, 10$. Other parameters: $m = 1, \epsilon = \hbar = 0.1, v = 1$.

itself is a very small fraction of the actual eigenenergies. The inset shows the relative error at $\beta = 1$ if we include higher-order corrections in $\beta$, comparing the eigenvalues of $\hat{H}_{\text{Gibbs}}$ (including corrections up to first- and second-order in $\beta$), with the energies obtained by calculating the expectation value of the similarly corrected $\hat{H}_{\text{Gibbs}}$ w.r.t. the eigenstates of $\mathcal{W}$. It is clear that the inclusion of higher-order terms reduces the relative error by orders of magnitude. This highlights the difference between the current approach and semiclassical approaches – as also follows from the fact that both corrections are of the same order in $\epsilon$: even though tunneling states are inherently non-classical and level splittings are hard to obtain using semiclassical approaches, these are already accurately reproduced from the zeroth-order approximation to $\hat{H}_{\text{Gibbs}}$ using the eigenstates of $\mathcal{W}$. Higher-order corrections in $\beta$ then only serve to increase the numerical accuracy but are not necessary to obtain a qualitative correspondence.

If we would interpret this in the language of quantum mechanics, such an accurate representation implies that the lowest symmetric and antisymmetric tunneling states are entangled. In the Fock basis of localized left- and right orbitals, these states read

$$|\psi_{\pm}\rangle \approx \sqrt{\frac{1}{2}} \left( |0\rangle_L |1\rangle_R \pm |0\rangle_R |1\rangle_L \right), \tag{76}$$

where $|0\rangle_{L,R}$ and $|1\rangle_{L,R}$ are the vacuum (no-particle) and the one-particle states corresponding to the left and right orbitals respectively. Both states are obviously maximally entangled. It is remarkable that such states are built into the classical Gibbs ensemble, and it can then be expected that entangled states with multiple particles can be equally accurately reproduced from the classical Gibbs ensemble.

The number of minima in the potential can be systematically increased to consider, e.g., a periodic lattice. Taking a potential of the form

$$V(x) = V_0(1 - \cos(x)) + V_{\text{conf}}(x), \qquad V_{\text{conf}}(x) = V_c \left( \frac{x}{40\pi} \right)^{10}, \tag{77}$$

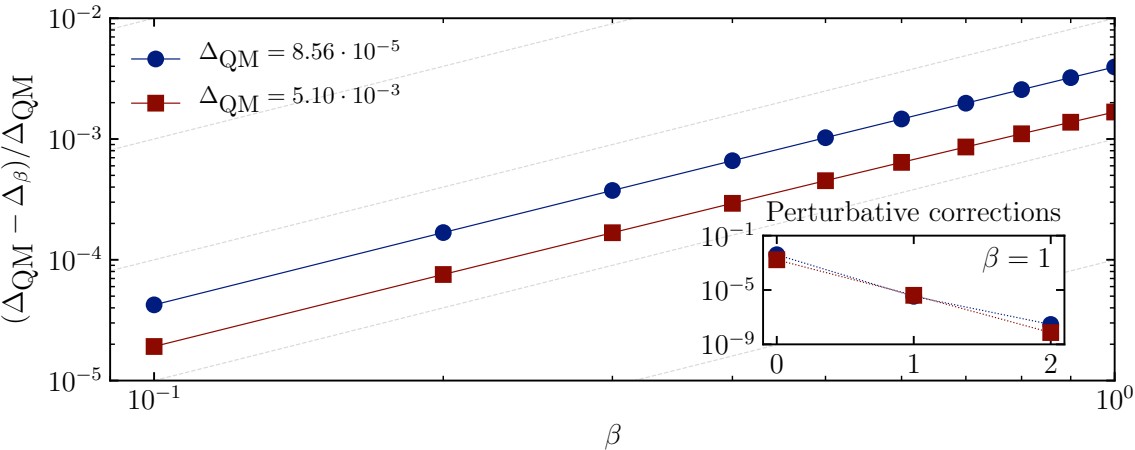

Figure 6: **Relative difference of the classical energy level splitting in a double well potential with respect to the quantum level splitting for the Gibbs temperature with varying $\beta$.** The blue circles correspond to the two lowest tunneling states and the red squares correspond to the second pair of tunneling states. In both lines the relative difference scales approximately as $\beta^2$ (gray lines). The inset shows the relative difference at $\beta = 1$ for the level splittings in $\hat{H}$ (0) and $\hat{H}_{\text{Gibbs}}$ including first (1) and second-order (2) corrections in $\beta$ as compared with those obtained from $\mathcal{W}$. The parameters are the same as in Fig. 5.

with $V_{0,c}$ constants, and where the last term represents an overall confining term helping to avoid dealing with boundary conditions and consider $x \in [-40\pi, 40\pi]$. In order to enhance dispersion, where tunneling again plays a crucial role, we choose a slightly smaller mass $m = 0.5$ while keeping $\epsilon = \hbar = 1$ and $\beta = 0.1$. In Fig. 7 we show the energy dispersions of the lowest band for eigenstates of both the quantum Hamiltonian and the classical Gibbs ensemble. The two lines are again visually indistinguishable, with errors on the order of $10^{-5}$.

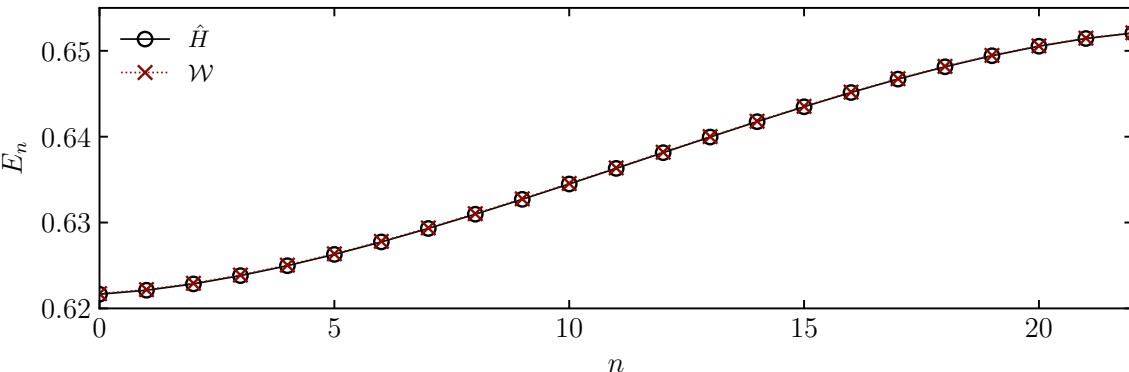

Figure 7: **Quantum and classical dispersion relations for a particle in a periodic potential** (77). The two lines representing the quantum and classical spectra are visually indistinguishable. The parameters are $V_0 = 1$, $V_c = 2$, $m = 0.5$, $\epsilon = \hbar = 1$, and $\beta = 0.1$

## 5.4 Two-dimensional coupled oscillator.

We now move to two-dimensional systems with a four-dimensional phase space, which we will take as $(x, p_x, y, p_y)$. For Hamiltonian evolution with

$$H(x, p_x, y, p_y) = \frac{p_x^2}{2m} + \frac{p_y^2}{2m} + V(x, y), \tag{78}$$

the previously obtained canonical distribution (49) at inverse temperature $\beta$ readily extends to

$$\mathcal{W}(x + \xi_x/2, y + \xi_y/2; x - \xi_x/2, y - \xi_y/2) = \frac{1}{Z_{xy}} \exp\left[-\frac{m}{2\beta\epsilon^2}(\xi_x^2 + \xi_y^2) - \beta V(x, y)\right], \tag{79}$$

with $Z_{xy} = \int dx \int dy \exp[-\beta V(x, y)]$ and where we can now define $(x_1, x_2) = (x + \xi_x/2, x - \xi_x/2)$ and $(y_1, y_2) = (y + \xi_y/2, y - \xi_y/2)$.

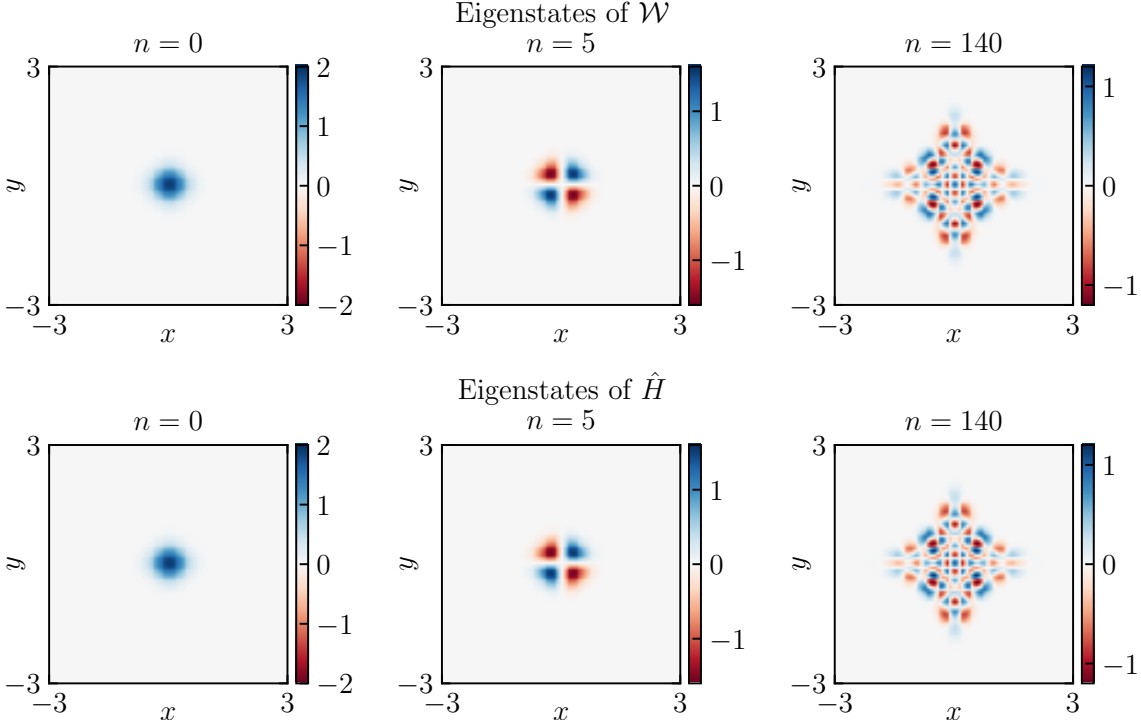

Figure 8: **Comparison of the classical eigenstates obtained from the Gibbs distribution and quantum eigenstates obtained solving the stationary Schrödinger equation for a two-dimensional potential.** Inverse temperature $\beta = 1$, potential corresponding to Eq. (80) with $m = \omega = 1$, $\delta = 0.1$, $\nu = 20$ and $\epsilon = \hbar = 0.1$.

As an example, we can consider an asymmetric coupled two-dimensional oscillator

$$V(x, y) = \frac{1}{2}m(\omega + \delta)^2 x^2 + \frac{1}{2}m(\omega - \delta)^2 y^2 + \frac{\nu}{4}x^2 y^2, \tag{80}$$

where the asymmetry is tuned by $\delta$ and the coupling by $\nu$. In Fig. 8, we again compare the eigenstates of $\mathcal{W}(x_1, y_1; x_2, y_2)$ with those of the quantum mechanical Hamiltonian, where we again choose $\beta = 1$ for $m = 1$ and $\epsilon = \hbar = 0.1$. The same qualitative behaviour as for one-dimensional systems can be observed, not just at low-lying states, but also at higher excited chaotic states. We compare the ground state ($n = 0$), the fifth excited state ($n = 5$)

and the 140th excited state ($n = 140$), where the latter is representative for general higher-energy states. The correspondence between the two is again excellent. Note that, while the effect of the coupling term on the ground and low-lying states might be small, it is crucial in the higher-energy states such as $n = 140$. The region where the wave function is non-zero corresponds to the classically-allowed region for a particle with a given energy, and the interaction is evidenced by the deformation of the edges of this region, where an ellipse would be obtained for two non-interacting oscillators with $\nu = 0$.

Interestingly, the same Hamiltonian (78) can be interpreted as describing a system of two particles in one dimension, where the non-linearity represents an interaction term. If the potential is symmetric under permutations of $x$ and $y$ (i.e. $\delta = 0$), this Hamiltonian describes two identical particles. From symmetry considerations, we can then classify all eigenstates according to their exchange symmetry, and we observe that the classical Gibbs distribution reproduces both. In particular, in the limit $\delta = 0$ the $n = 0$ state shown in Fig. 8 is even under the exchange $x \leftrightarrow y$ and hence bosonic, while the $n = 5$ is odd and thus fermionic. While we focus on stationary states, it is clear that if the exchange symmetry is preserved under dynamics then the even and odd states cannot mix and evolve independently from each other, as in quantum mechanics. This implies that in the state language representation classical dynamics can also be expressed through independent evolution of bosonic and fermionic wave functions.

## 5.5 Magnetic fields.

We now consider two-dimensional systems with an additional magnetic field, following Section 4.3. Similar to the harmonic oscillator, an analytic solution is possible for a uniform field $B$ perpendicular to the plane in the absence of any potential term $V(x, y)$. Quantum mechanically, this situation gives rise to the famous quantized Landau levels, $E_n = \hbar\omega(n + 1/2), n \in \mathbb{N}$, with $\omega = qB/mc$ the so-called cyclotron frequency.

In this first set-up, the saddle-point approximation is exact, and we can recover the Landau states directly from the classical Boltzmann distribution. Considering a vector potential $\vec{A} = (-By, 0, 0)$, the Boltzmann distribution is given by

$$P(x, p_x, y, p_y) = \frac{1}{Z} \exp\left[-\frac{\beta}{2m}\left(p_x + \frac{qBy}{c}\right)^2\right] \exp\left[-\frac{\beta}{2m}p_y^2\right], \tag{81}$$

and the resulting quasi-density matrix can be analytically obtained from Gaussian integration as

$$\mathcal{W}(x_1, y_1; x_2, y_2) = \frac{1}{Z} \exp\left[-\frac{m(x_2 - x_1)^2}{2\epsilon^2\beta}\right] \exp\left[-\frac{m(y_2 - y_1)^2}{2\epsilon^2\beta}\right]$$
$$\times \exp\left[-i\frac{qB}{\epsilon c}(y_2 + y_1)(x_2 - x_1)\right]. \tag{82}$$

The vector potential clearly results in a quasi-density matrix that is no longer real. As shown in Appendix E, Eq. (82) can be explicitly diagonalized by combining the Fourier transform with an identity by Weiner, resulting in

$$\mathcal{W}(x_1, y_1; x_2, y_2) = \sum_{n=0}^{\infty} w_n \int dk\, \psi_{n,k}^*(x_1, y_1)\psi_{n,k}(x_2, y_2), \tag{83}$$

where the eigenstates are labelled by the continuous momenum along the $x$-direction $k$ and a discrete index $n$. These states are given by the Landau states

$$\psi_{n,k}(x, y) = \frac{1}{\sqrt{2\pi}}\psi_n\left(y - \frac{\epsilon}{m\omega}k\right)e^{ikx}, \tag{84}$$

where $\psi_n(x)$ are the eigenstates of the harmonic oscillator (66) with natural frequency given by the cyclotron frequency $\omega$. The eigenvalues only depend on $n$ and are found as

$$w_n = \frac{1}{Z}\left(\frac{1-\beta\epsilon\omega/2}{1+\beta\epsilon\omega/2}\right)^n \sqrt{\frac{\pi\beta\epsilon\omega}{(1+\beta\epsilon\omega/2)^2}}. \tag{85}$$

In the high-temperature limit $\beta\epsilon\omega \ll 1$ the weights $w_n$ are all positive, returning the quantum Gibbs ensemble:

$$w_n \approx \frac{1}{\mathcal{Z}}\exp[-\beta\epsilon\omega(n+1/2)]. \tag{86}$$

As the temperature decreases (but remains larger than $\epsilon\omega/2$) the weights remain equidistant. As with the harmonic oscillator, it is possible to formally define a quantum temperature $\beta_q$ according to Eq. (67), which immediately follows from $\mathrm{arctanh}(x) = \frac{1}{2}\log\frac{1+x}{1-x}$. At $\beta^{-1} = \epsilon\omega/2$ the quantum temperature reaches zero, corresponding to the occupancy of a single (highly degenerate) Landau level. Further increasing $\beta$ again results in oscillatory behavior in the eigenvalues $w_n$, with the effective temperature $\tilde{\beta}_q$ given by Eq. (68).

Numerically, the correspondence between the quantum eigenstates and the classical eigenstates remains accurate in the presence of a vector potential even if the system is no longer exactly solvable. This is illustrated in Fig. 9, where we consider a vector potential

$$\vec{A}(x,y) = \frac{\Phi}{2\pi(x^2+y^2+\delta^2)}(-y,x,0), \tag{87}$$

corresponding to a radially decaying magnetic field pointing in the $z$-direction with an additional confining potential

$$V(x,y) = m\omega^2(x^2+y^2-1)^2. \tag{88}$$

Selected eigenstates of the quasi-density matrix $\mathcal{W}$ are given in Fig. 9, and compared with the eigenstates of the quantum Hamiltonian (63) (without $\beta$ correction). We consider the ground state ($n = 0$) and an excited state ($n = 18$). Since the eigenstates will generally be complex, we compare both the amplitude and the phase of the eigenstates. The correspondence between both is again clear. We observe that the classical approach is able to reproduce the nontrivial complex (Berry) phase of the quantum eigenstates, which we checked survives even in the Aharonov-Bohm geometry.

# 6 Bohigas-Giannoni-Schmit conjecture for classical systems

One of the important breakthroughs in our understanding of quantum chaos came through the Bohigas-Giannoni-Schmit (BGS) conjecture [29], postulating that in chaotic Hamiltonians the eigenvalue spectrum obeys random matrix statistics. This conjecture essentially extends an earlier conjecture by M. Berry [30], who proposed that stationary states in chaotic billiards can be well approximated by random linear combinations of plane waves. This can be contrasted with "generic" integrable, i.e. non-chaotic, systems, where the eigenvalue spectrum is expected to exhibit Poissonian statistics following the Berry-Tabor conjecture [31]. Level statistics have since become one of the predominant diagnostics of quantum chaos. Still, there is a longstanding and still not fully resolved problem of reconciling this notion of quantum chaos with that of classical chaos, defined through Lyapunov exponents of individual trajectories [32–39]. Interestingly, and as we discuss here, these conjectures apply to classical systems as well.

As an illustration, we will take the classic example of a Sinai-like billiard, with the potential shown in Fig. 10. This potential is constructed from a series of smoothened step functions

$$\mathrm{St}(x) = 4(\tanh(10x)-1),$$

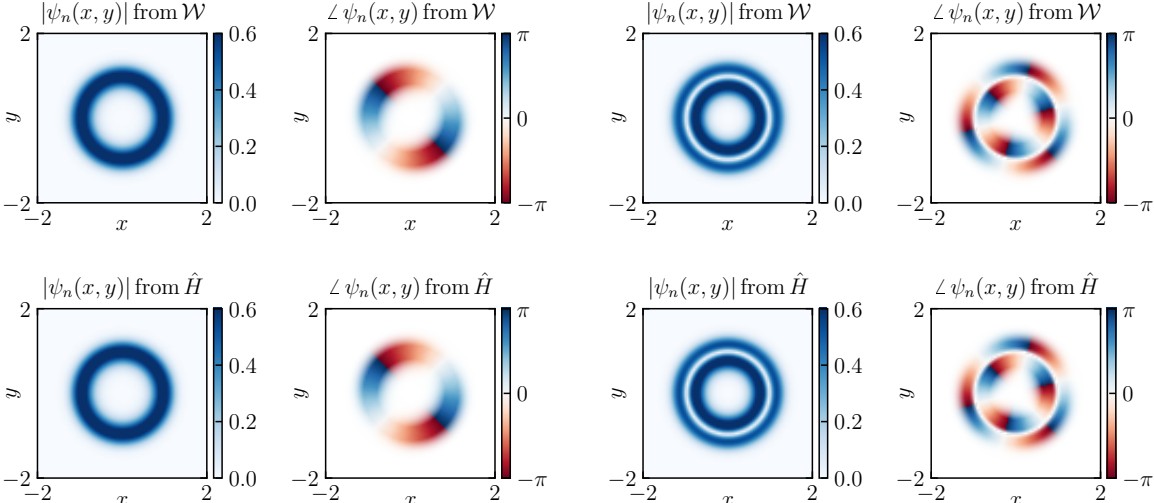

Figure 9: **Comparison of absolute value $|\psi_n|$ and phase $\angle\psi_n$ of classical eigenstates obtained from the Gibbs distribution (top row) and quantum eigenstates obtained solving the stationary Schrödinger equation (bottom row) for a two-dimensional potential in the presence of a magnetic field.** Inverse temperature $\beta = 0.5$, potential corresponding to Eqs. (87) and (88) with $m = \omega = 1$, $\delta = 2$, $\Phi = \pi$, and $\epsilon = \hbar = 0.1$. Transparency for the phase is set by the absolute value in order to avoid numerical noise where the absolute value is small.

and we will consider one specific profile given by

$$V(x, y) = \mathrm{St}\big((\mathrm{St}(-y + 0.3) + \mathrm{St}(x - r_2) + \mathrm{St}(r_1^2 - x^2 - y^2) + \mathrm{St}(-x + 0.3)$$
$$+ \mathrm{St}(y - \tan(2\pi/5)x) + \mathrm{St}(y - r_2))\big) - 4 + \frac{((x - r_2/2)^2 + (y - r_2/2)^2)^2}{16}. \quad (89)$$

The last term represents an overall weak confining potential to avoid dealing with non-analytic boundary conditions. The specific choice of the parameters in the potential is unimportant for the subsequent analysis and is only given for completeness.

We analyze a particle of a unit mass in this potential, setting $\epsilon = 0.2$ and $\beta = 0.3$. Such a relatively small value of $\epsilon$ is necessary to have enough confined energy levels to be able to extract meaningful statistics. Furthermore, we change the inner radius of this potential $r_1$ incrementally between $r_1^{\min} = 1.5$ and $r_1^{\max} = 1.78$ in steps of 0.02 and similarly vary $r_2$ between $r_2 = r_1 + 1.5$ and $r_2 = r_1 + 1.78$ with the same step size, leading to $15 \times 15 = 225$ realizations of this potential. For each realization we analyze the spectrum of the Gibbs distribution $\mathcal{W}$ and choose 900 eigenvalues $w_n$ between $n = 100$ and $n = 1000$, with $n = 0$ corresponding to the highest weight state (ground state). For this low value of $\beta$ the eigenvalues are positive, so we can define an eigenspectrum through the logarithm $\lambda_n = -\log(w_n)$, corresponding to the quantum eigenenergy in the limit $\beta \to 0$. We verified that one can directly analyze the weights $w_n$ without affecting the results.

To check whether the spectrum satisfies the BGS conjecture, i.e. exhibits random matrix statistics described by a Gaussian Orthogonal Ensemble (GOE), we take the measure proposed in Ref. [40] (see also Ref. [41]) analyzing the distribution of the ratio between consequent eigenvalue spacings:

$$r_n = \frac{\lambda_{n+1} - \lambda_n}{\lambda_{n+2} - \lambda_{n+1}} = \frac{\log(w_n/w_{n+1})}{\log(w_{n+1}/w_{n+2})}. \quad (90)$$

For the GOE distribution the probability of this ratio is well described by the analytic expres-

sion [40]:

$$P(r) \approx \frac{27}{8} \frac{r + r^2}{(1 + r + r^2)^{5/2}} + \frac{0.233378}{(1+r)^2} \left[ \left( r + \frac{1}{r} \right)^{-1} - \frac{2\pi - 4}{4 - \pi} \left( r + \frac{1}{r} \right)^{-2} \right], \quad (91)$$

whereas a Poissonian distribution results in

$$P(r) = \frac{1}{(1+r)^2}. \quad (92)$$

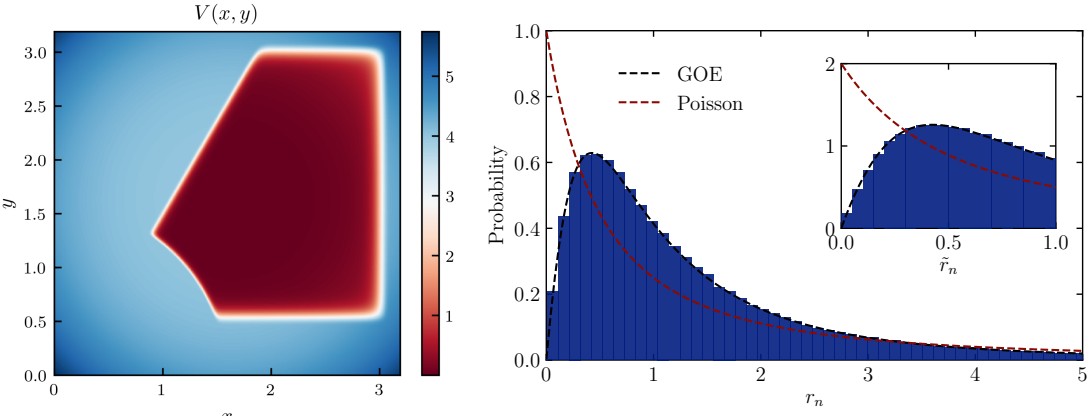

Figure 10: **Distribution of the ratio of consecutive level spacings**. **Left:** The chaotic two-dimensional potential representing a series of smoothened step functions in an additional weak confining potential. **Right:** Histogram of the ratio of the consequent difference of the (logs of) eigenvalues (see Eq. (90)) of the canonical distribution $\mathcal{W}$ for a particle in this potential. The black dashed line is the analytic approximation to the GOE distribution taken from Ref. [40] and the red dashed line is the expected result for Poissonian statistics (92). Inset details the distribution of $\tilde{r}_n = \min(r_n, 1/r_n)$.

In the right panel of Fig. 10 we plot the numerically computed histogram of ratios $r_n$ for the particle in the potential (89) and for comparison show the approximate analytic expression for the distribution expected for the GOE ensemble and the Poissonian results. The inset illustrates the distribution of another commonly used measure of level statistics: $P(\tilde{r}) = 2P(r)\theta(1-r)$, where $\tilde{r}_n = \min(r_n, 1/r_n)$ [40]. In both cases the correspondence with the GOE is nearly perfect, fully supporting the BGS conjecture for classical eigenstates in chaotic potentials.

Finally, note that Poissonian level statistics can also be expected to appear in classically integrable systems, validating the Berry-Tabor conjecture. E.g. for a classical two-dimensional harmonic oscillator with mismatched frequencies, the classical spectrum exactly corresponds to that of the quantum Hamiltonian, which is known to exhibit Poissonian level statistics. In Fig. 11, we numerically verify this for an elliptic confining potential,

$$V(x, y) = \text{St}\left( (x/r_1)^2 + (y/r_2)^2 - 1 \right). \quad (93)$$

We again choose $\epsilon = 0.2$ and $\beta = 0.3$. For the sampling of eigenvalues, we vary $r_1$ from 1.22 to 1.50 and $r_2$ from 0.91 to 1.19 in steps of 0.02, leading to 225 realizations of this potential. These values are generic, and chosen so as to avoid any spurious degeneracies. Performing a similar sampling as for Fig. 10, the level statistics are clearly Poissonian and indicate a non-ergodic potential.

This finding is in a way remarkable. It illustrates how the classical Gibbs distribution, a stationary state, already knows whether the system is chaotic or not – in line with the quantum

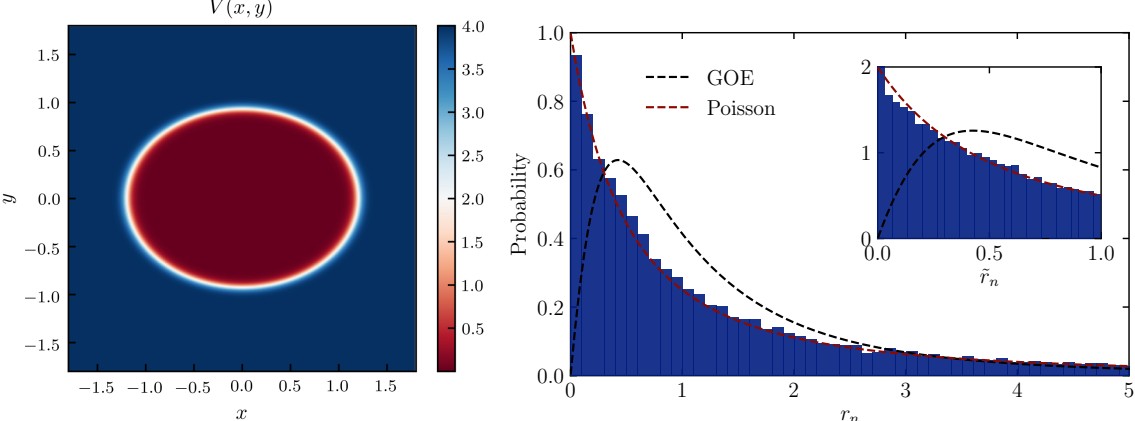

Figure 11: **Distribution of the ratio of consecutive level spacings**. **Left:** The symmetric potential representing a smoothened elliptic confining potential (93). **Right:** Histogram of the ratio of the consequent difference of the (logs of) eigenvalues (see Eq. (90)) of the canonical distribution $\mathcal{W}$ for a particle in this potential. See also 10.

definition of chaos. This analysis does not require us to analyze any dynamical response, compute Lyapunov exponents, or solve any equations of motion to understand whether the system is chaotic or not. Moreover, we do not even need to use the fact that $x$ and $p$ are canonically conjugate variables. This statement suggests the existence of a much closer relation between thermodynamics and dynamics even at the level of purely classical mechanics.

Finally let us comment that if, instead of analyzing the statistics of $w_n$, we would calculate the statistics of exact classical mean energies $E_n = \langle \psi_n | \hat{H} | \psi_n \rangle$ defined through the classical eigenstates $|\psi_n\rangle$, we would not recover the RMT statistics. While for this particular system $E_n$ and $-\log(w_n)/\beta$ are in a very good correspondence in the entire analyzed range of $n$, there are small random fluctuations in the energy spectrum calculated using the expectation values that suffice to destroy the level repulsion between $E_n$.

## 7 Conclusions and Outlook

We highlighted how key concepts in classical mechanics can be reformulated in the language of Hermitian operators and states more familiar from quantum mechanics. This language naturally follows from applying the inverse Wigner-Weyl transform to the classical probability distribution $P(x, p)$, mapping it to a function $\mathcal{W}(x_1, x_2) = \mathcal{W}^*(x_2, x_1)$ playing the role of the density matrix in the language of quantum mechanics. This function can in turn be diagonalized, with its eigenfunctions $\psi_n(x)$ playing a role similar to quantum wave functions. The commutation relation between canonical operators $\hat{x}$ and $\hat{p}$ naturally emerges from this mapping to a quasi-density matrix, even though such commutators are generally associated with the classical Poisson bracket determining the dynamics of the phase-space variables. We then showed that the correspondence with quantum mechanics is particularly striking if $P(x, p)$ is described by the classical Gibbs ensemble. Then the corresponding classical eigenstates exactly match quantum eigenstates in the limit of high temperature. Surprisingly, at least in many situations, this correspondence remains highly accurate even if all the parameters remain of the order of unity. In particular, we were able to accurately describe both ground and excited wave functions in a nonlinear quartic potential, a double-well potential containing tunneling states, a band structure in a periodic one-dimensional potential, and both low-energy states and highly excited states in a two-dimensional nonlinear (chaotic) potential and a two-

dimensional potential with magnetic field. The correspondence was analytically shown to be exact at all temperatures for the harmonic oscillator and for Landau levels.

Not only do these classical eigenstates generally correspond to quantum states at high temperatures, the eigenvalues of the quasi-density matrix, commonly interpreted as the probabilities to occupy the eigenstates, return the expected quantum Gibbs/Boltzmann factors. As the temperature is lowered, or more generally as the classical distribution becomes narrower, the classical weights of eigenstates start to acquire negative values and can now be interpreted only as quasi-probabilities. This is exactly dual to the Wigner function, where the phase space distribution following from a quantum density matrix can take non-positive values. Interestingly, there is always a minimum temperature set by the quantum uncertainty relation where the representation of $\mathcal{W}(x_1, x_2)$ and hence $P(x, p)$ through the eigenstates becomes 'maximally discrete', i.e. contains the fewest number of non-negligible components. This can be quantified through various measures, where we here consider an extension of Rényi entropies to negative weights. For example, given an oscillator at a temperature $T = \epsilon\omega/2$ ($\epsilon$ is a free parameter in classical mechanics, playing the role of $\hbar$) only a single (ground) state is occupied. At both higher and lower temperatures an increasing number of states are occupied. Interestingly, the classical partition function of oscillators with $T > \epsilon\omega/2$ maps to the quantum partition function of bosons with energy scale $\epsilon\omega$, while the classical partition function of oscillators with $T < \epsilon\omega/2$ maps exactly to the partition function of free fermions with the same energy scale $\epsilon\omega$. Whether this is a simple coincidence or if there is a deeper underlying reason remains to be understood.

In this work we focused only on unbounded potentials and hence did not emphasize the role of boundary conditions: obviously both classical and quantum probabilities have to vanish deep in the energetically-forbidden region. It is clear that understanding other, e.g. periodic, boundary conditions should prove to be very interesting. For example, if we consider a particle in a central potential the classical Gibbs probability distribution has to be a periodic function of the angular variable. This implies that the classical states can be both periodic and anti-periodic functions of the angle, with a striking similarity of these two possibilities to integer and half-integer spin. We left analyzing such possibilities to a future work. Similarly, the extension to multiple classical particles and the resulting particle statistics should also be highly interesting. In the presented mapping the number of particles play no role, so at least at high temperatures the many-particle stationary states should satisfy the correct Schrödinger equation. Many more questions such as adiabatic continuation, the manifestation of the discreteness of stationary states, the relaxation of interacting systems to equilibrium, linear response theory,... were left out of the present work, offering various directions for future research. One of the cornerstones of our current understanding of quantum thermalization is given by the eigenstate thermalization hypothesis [35], relating matrix elements of quantum states to thermal distributions, and the connection between quantum states and classical distributions could also be revisited in this context. Classically, stationary distributions distinct from the thermal and microcanonical ones are guaranteed to exist as time-averaged KAM trajectories [42–44], and it would similarly be interesting to check the correspondence between the eigenstates following from this classical distribution and the quantum Hamiltonian. From our discussion it should be clear that there is a close connection between the classical Liouville equation and the quantum Schrödinger equation, so it is inevitable that various quantum dynamical phenomena are encoded in the operator-state representation of classical systems.

Finally let us point out that the ideas presented here can go beyond classical mechanics. Given a classical probability distribution in position space, it is always possible to introduce momentum as an auxiliary degree of freedom, as is often done in e.g. annealing problems. The construction shown here can be seen as a way to map such a continuous to a discrete probability distribution, in the same way that stationary quantum mechanics presents a dis-

crete representation of the Gibbs distribution. Similarly, given a continuous distribution of two (or any even number of) variables, in this way one can always represent this probability distribution as a discrete sum of eigenstates depending on a single variable. This construction can be viewed as an effective dimensional reduction of the original distribution. One can possibly continue this procedure of dimensional reduction in a system with more variables, reducing their number by a factor of two at each step. The parameter $\epsilon$ (or equivalently $\hbar$) can then be chosen for convenience, e.g. minimizing the number of discrete components in such representations.

## Acknowledgements

We gratefully acknowledge Daniel Arovas, Jan Behrends, Denys Bondar, Anatoly Dymarsky, Steven Girvin, Pankaj Mehta, Marcos Rigol, Dries Sels, and Jonathan Wurtz for useful comments and discussions. We additionally acknowledge Pankaj Mehta for sharing his original derivation of the classical Landau levels. We are especially grateful to Sir Michael Berry for his valuable comments and suggestions, leading to several improvements and additional results in the revised manuscript.

**Funding information** P.W.C. gratefully acknowledges support from EPSRC Grant No. EP/P034616/1. Work of A.P. was supported by NSF DMR-1813499 and AFOSR FA9550-16- 1-0334.

## A  Dynamics & General Stationary States

Follwing Section 3.1, the equation of motion for $\mathcal{W}$ can be written as

$$i\epsilon \frac{\partial}{\partial t}\mathcal{W} = \mathcal{L}[\mathcal{W}], \qquad \mathcal{L} = -\frac{\epsilon^2}{2m}\left[\frac{\partial^2}{\partial x_2^2} - \frac{\partial^2}{\partial x_1^2}\right] - (x_1 - x_2)V'\left(\frac{x_1 + x_2}{2}\right). \qquad (94)$$

Introducing a discrete basis of eigenoperators of $\mathcal{L}$, the coupled differential equations of classical mechanics will here lead to a solution of $\mathcal{W}$ described by dephasing eigenoperators of the superoperator $\mathcal{L}$, familiar from quantum mechanics.

Denoting the complete set of orthonormal eigenoperators of $\mathcal{L}$ as $\mathcal{O}_\alpha(x_1, x_2)$ with eigenvalues $\lambda_\alpha$, the classical dynamics is given by

$$\mathcal{W}(x_1, x_2, t) = \sum_\alpha e^{-\frac{i}{\epsilon}\lambda_\alpha t}(\mathcal{O}_\alpha|\mathcal{W})\mathcal{O}_\alpha(x_1, x_2), \qquad (95)$$

where the expansion coefficients of $\mathcal{W}$ are given by

$$(\mathcal{O}_\alpha|\mathcal{W}) = \int dx_1 \int dx_2 \, \mathcal{O}_\alpha^*(x_1, x_2)\mathcal{W}(x_1, x_2, t=0), \qquad (\mathcal{O}_\alpha|\mathcal{O}_\beta) = \delta_{\alpha\beta}. \qquad (96)$$

By making use of the fact that $\mathcal{L}$ is antisymmetric under exchange of $x_1 \leftrightarrow x_2$, it follows that nonzero-eigenvalue eigenoperators of $\mathcal{L}$ arise in pairs, where an eigenoperator $\mathcal{O}_\alpha(x_1, x_2)$ with nonzero eigenvalue $\lambda_\alpha$ leads to another eigenoperator $\mathcal{O}_\alpha(x_2, x_1)$ with eigenvalue $-\lambda_\alpha$. This statement can easily be checked in a known limit: assuming the phase space distribution is smooth enough at all times such that we can approximate $\mathcal{L}[\mathcal{W}] = [\mathcal{W}, \hat{H}]$, or that the potential is close to harmonic, the eigenoperators of $\mathcal{L}$ are simply products of eigenstates

of the Hamiltonian. Labeling the eigenstates of $\hat{H}$ as $\psi_n(x)$ with eigenvalue $E_n$, the corresponding eigenoperators of $\mathcal{L}$ are given by $\mathcal{O}_{nm}(x_1, x_2) = \psi_m(x_1)\psi_n(x_2)$ with eigenvalues $\lambda_{nm} = E_n - E_m$. These clearly arise in pairs, since $\lambda_{mn} = E_m - E_n = -\lambda_{nm}$ for the eigenoperator $\psi_m(x_2)\psi_n(x_1)$. Stationary states, which can also be obtained from the long-time average of $\mathcal{W}(x_1, x_2, t)$, here correspond to the zero-eigenvalue eigenoperators of $\mathcal{L}$ and reduce to the diagonal states $\psi_n(x_1)\psi_n(x_2)$ in this limit.

Stationary distributions are necessarily eigenoperators of $\mathcal{L}$ with eigenvalue zero. It is possible to expand $\mathcal{W}$ in an arbitrary basis as

$$\mathcal{W}(x_1, x_2) = \sum_{\alpha\beta} \mathcal{W}_{\alpha\beta} \psi_\alpha^*(x_1)\psi_\beta(x_2), \tag{97}$$

where $\{\psi_\alpha(x)\}$ are some complete set of orthonormal wave functions, which could be e.g. eigenstates of a quantum Hamiltonian $\hat{H}$. Plugging this expansion into Eq. (21), multiplying both parts of this equation by $\psi_\gamma(x_1)\psi_\delta^*(x_2)$ and integrating over $x_1$ and $x_2$, we find the exact equation for the matrix elements of the stationary $\mathcal{W}$

$$\sum_{\alpha\beta} \mathcal{L}_{\gamma\delta}^{\alpha\beta} \mathcal{W}_{\alpha\beta} = 0, \tag{98}$$

where $\mathcal{L}$ is the superoperator with entries

$$\mathcal{L}_{\gamma\delta}^{\alpha\beta} = \int dx_1 \int dx_2 \, \psi_\gamma(x_1)\psi_\delta^*(x_2) \left[ -\frac{\epsilon^2}{2m} \left( \frac{\partial^2}{\partial x_2^2} - \frac{\partial^2}{\partial x_1^2} \right) - \xi V'(x) \right] \psi_\alpha^*(x_1)\psi_\beta(x_2). \tag{99}$$

In the limit of sufficiently small $\epsilon$ this equation clearly reduces to the matrix form of the stationary von Neumann's equation

$$[\mathcal{H}, \mathcal{W}]_{\gamma\delta} = 0, \quad \mathcal{H}_{\alpha\beta} = \langle \psi_\alpha | \hat{H} | \psi_\beta \rangle \equiv \int dx \, \psi_\alpha^*(x)\hat{H}\psi_\beta(x). \tag{100}$$

In general, solutions of Eq. (98) are highly degenerate and the number of solutions generally corresponds to the number of eigenstates of the Hamiltonian, reflecting how each state results in a stationary distribution. However, different stationary distributions $\mathcal{W}(x_1, x_2)$ are not expected to commute and different stationary contributions will generally have different eigenstates, such that the set of stationary eigenstates is not uniquely defined.

## B  Microcanonical ensemble

In one-dimensional systems, assuming that there are no spatially disconnected regions in phase space, any stationary distribution can be represented as a statistical mixture of microcanonical distributions [45]:

$$P(x, p) = \int dE \, \rho(E) P_{\text{mc}}(x, p; E), \tag{101}$$

where $\rho(E)$ is the energy distribution function. The microcanonical distributions are characterized by an equal probability of occupying phase space points on the constant energy surface as

$$P_{\text{mc}}(x, p; E) = \frac{1}{Z} \delta\left( E - \frac{p^2}{2m} - V(x) \right), \qquad Z = \int dx \int dp \, \delta\left( E - \frac{p^2}{2m} - V(x) \right). \tag{102}$$

Since $P_{\mathrm{mc}}(x,p;E)$ is a function of the Hamiltonian of the system $H(x,p)$ it automatically satisfies $\{H,P\}=0$, such that it is necessarily stationary. Such microcanonical distributions naturally arise when considering long-time averages of classical trajectories [45].

In the following, we will analytically present results for systems with a linear potential and a quadratic potential (harmonic oscillator). In these cases the operator $\mathcal{L}$ exactly reduces to the commutator with the Hamiltonian for an arbitrary $\epsilon$. Therefore all zero-eigenvalue eigenoperators of $\mathcal{L}$ necessarily commute with the Hamiltonian, sharing the same eigenstates, which does not hold for more general potentials. Still, it is instructive to consider these simple examples to understand the behavior of the eigenvalues.

**Linear potential.**

For a linear potential $V(x)=\alpha x$, the eigenstates of the quantum Hamiltonian can be expressed as Airy functions. As shown in Appendix C, the resulting quasi-density matrix can be obtained by combining two identities for the Airy functions, such that the transform of Eq. (102) for $V(x)=\alpha x$ can be explicitly written in its diagonal form as

$$\mathcal{W}_{\mathrm{mc}}(x_1,x_2;E)=\frac{2\pi\epsilon}{Z}\frac{2^{2/3}}{\alpha^2\lambda^3}\int d\tilde{E}\,\mathrm{Ai}\left[\frac{2^{2/3}}{\alpha\lambda}(E-\tilde{E})\right]\mathrm{Ai}\left(\frac{x_1-\tilde{E}/\alpha}{\lambda}\right)\mathrm{Ai}\left(\frac{x_2-\tilde{E}/\alpha}{\lambda}\right),\quad(103)$$

where we have introduced the customary length scale $\lambda^3=\epsilon^2/(2m\alpha)$ and the Airy functions are the eigenstates of the Hamiltonian with linear potential[8]. For an eigenstate of the Hamiltonian with energy $\tilde{E}$, the corresponding eigenvalue of the quasi-density matrix of the microcanonical ensemble with energy $E$ is given by

$$w_{\tilde{E}}=\frac{2^{2/3}}{\lambda\alpha}\mathrm{Ai}\left[\frac{2^{2/3}}{\lambda\alpha}(E-\tilde{E})\right]\qquad\text{with}\qquad\frac{2^{2/3}}{\lambda\alpha}\int d\tilde{E}\,\mathrm{Ai}\left[\frac{2^{2/3}}{\lambda\alpha}(E-\tilde{E})\right]=1.\quad(104)$$

Whereas it might be expected that a classical distribution with fixed energy $E$ only contains contributions from quantum states with a similar energy $\tilde{E}$, quite the opposite happens: for a microcanonical state with classical energy $E$ its quasi-density matrix contains contributions from almost all eigenstates of the Hamiltonian with quantum energies $\tilde{E}$, where the eigenvalue is determined by the Airy function of $E-\tilde{E}$. For states with $\tilde{E}\ll E$, the eigenvalue will be exponentially suppressed, whereas for states with $\tilde{E}\gg E$ the eigenvalues are highly oscillatory and only decaying as $(\tilde{E}-E)^{-1/4}$. Clearly, a large fraction of the latter eigenstates also have a negative eigenvalue, and the same oscillatory behaviour as for the Gaussian distribution (45) can be observed.

This distinction vanishes if we allow for sufficient uncertainty on the classical energy. Assuming a Gaussian uncertainty on the energy centered on $E=0$ in Eq. (101) as

$$\rho(E)=\frac{1}{\sqrt{2\pi}\sigma}\exp\left[-\frac{E^2}{2\sigma^2}\right],\quad(105)$$

the eigenstates of the corresponding $\mathcal{W}$ will remain unchanged (they do not explicitly depend on $E$), whereas the eigenvalues are now given by

$$w_{\tilde{E}}=\frac{2^{2/3}}{\lambda\alpha}\int dE\,\rho(E)\,\mathrm{Ai}\left[\frac{2^{2/3}}{\lambda\alpha}(E-\tilde{E})\right],\quad(106)$$

---

[8]Note that $Z$ is ill-defined since we did not impose boundary conditions for $x\to-\infty$ and the eigenstates are not normalizable. This can be solved by imposing boundary conditions and would also take care of the factor $2\pi\epsilon/Z$ in the normalization.

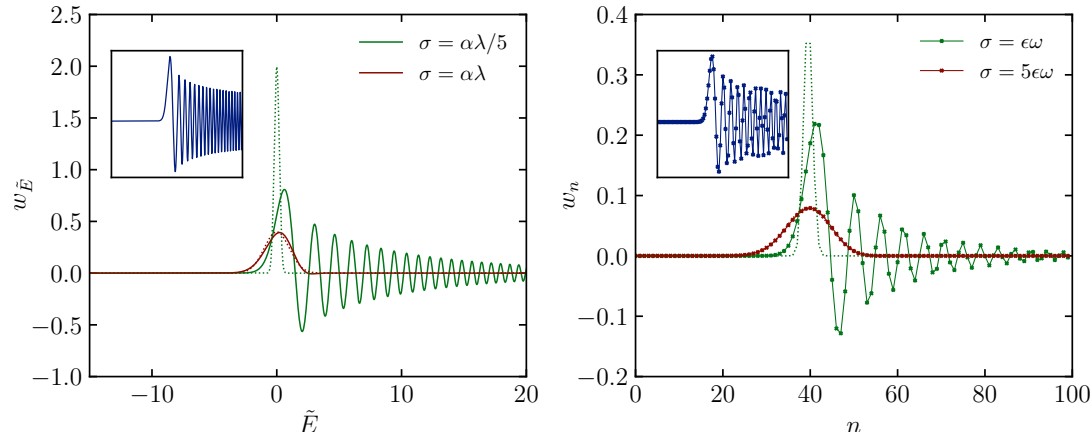

Figure 12: **The eigenvalues of a microcanonical distribution with Gaussian uncertainty on the classical energy reproduces the original distribution for sufficient width.** Left: linear potential, full line is averaged $w_{\tilde{E}}$, dashed line is $\rho(\tilde{E})$ as a Gaussian with width $\sigma$ centered on $E_{\mathrm{avg}} = E = 0$ and $\alpha\lambda = 1$. Right: harmonic potential, full line is averaged $w_n$, dashed line is $\rho((n+1/2)\epsilon\omega)\epsilon\omega$, with $\epsilon\omega = 1$ and the Gaussian with width $\sigma$ centered on $E = E_{\mathrm{avg}} = 4$. In both figures the inset details the eigenvalues without any uncertainty, where a connecting line is added to the right (discrete) eigenvalues to guide the eye.

i.e. the Airy transform of the probability distribution of the microcanonical energy. The resulting eigenvalues are presented in the left panel of Fig. 12 for a Gaussian distribution with different widths. For small $\sigma$ the resulting distribution still resembles the Airy function, albeit with a quicker decay, but for larger $\sigma \gtrsim \alpha\lambda$ all oscillations cancel out and we numerically obtain $w_{\tilde{E}} = \rho(\tilde{E})$: all negative eigenvalues have effectively been averaged out to zero and the quantum and classical states agree.

**Harmonic oscillator.**

The same derivation can be repeated for the harmonic oscillator with $V(x) = \frac{1}{2}m\omega^2 x^2$, where the eigenstates of the Hamiltonian are expressed in terms of Hermite polynomials $H_n$ as

$$\psi_n(x) = \frac{1}{\left(\pi\xi_0^2\right)^{1/4}}\frac{1}{\sqrt{2^n n!}}H_n(x/\xi_0)\exp\left[-\frac{x^2}{2\xi_0^2}\right], \qquad \xi_0^2 = \frac{\epsilon}{m\omega}, \tag{107}$$

satisfying the eigenvalue equation with a discrete eigenspectrum labeled by an index $n = 0, 1, 2, \ldots$

$$\left[-\frac{\epsilon^2}{2m}\frac{d^2}{dx^2} + \frac{1}{2}m\omega^2 x^2\right]\psi_n(x) = \epsilon\omega\left(n + \frac{1}{2}\right)\psi_n(x). \tag{108}$$

Using known properties of the Hermite and Laguerre polynomials, we show in Appendix C.2 that for the harmonic oscillator the transform of the microcanonical ensemble can be expanded as

$$\mathcal{W}_{\mathrm{mc}}(x_1, x_2; E) = \sum_{n=0}^{\infty} 2(-1)^n L_n\left(\frac{4E}{\epsilon\omega}\right)\exp\left(-\frac{2E}{\epsilon\omega}\right)\psi_n(x_1)\psi_n(x_2), \tag{109}$$

in which $L_n$ are the Laguerre polynomials. The eigenstates are again the eigenstates of the quantum Hamiltonian, now labeled with a discrete index $n$, where the corresponding eigen-

value for the microcanonical ensemble with energy $E$ is given by

$$w_n(E) = 2(-1)^n L_n\left(\frac{4E}{\epsilon\omega}\right)\exp\left(-\frac{2E}{\epsilon\omega}\right). \tag{110}$$

These exhibit the same qualitative behaviour as for the linear potential: given a microcanonical distribution with energy $E$, the eigenvalue distribution is peaked at the eigenstate of the Hamiltonian with the same energy. All other eigenvalues have strong positive and negative contributions, either exponentially decaying away from the peak at small $n$ and correspondingly $E_n = \epsilon\omega(n+1/2) < E$ or oscillating at $E_n > E$. Adding some uncertainty on the energy of the microcanonical state, all oscillations cancel out, and we end up with an eigenvalue distribution centered around the corresponding eigenstate, as illustrated in the right panel of Fig. 12. The distribution of eigenvalues is now the Laguerre transform of the energy distribution, and we numerically observe that for sufficiently large uncertainty

$$\int dE\rho(E)w_n(E) \to \epsilon\omega\rho\left[\epsilon\omega(n+1/2)\right]. \tag{111}$$

Note that the eigenvalues $w_n(E)$ are highly similar to the Wigner function in the position-momentum space corresponding to a single $n$-th level of a quantum harmonic oscillator, as also argued in Appendix C.2.

**General potentials.**

Before continuing to canonical potentials, we briefly discuss how these previous results extend to more general potentials. Starting from a microcanonical ensemble with fixed energy $E$ and potential $V(x)$, it is straightforward to find the function $\mathcal{W}_{\mathrm{mc}}(x_1, x_2)$ corresponding to the microcanonical distribution as

$$\mathcal{W}_{\mathrm{mc}}(x + \xi/2, x - \xi/2; E) = \frac{2}{Z}\frac{\theta(E - V(x))}{v_E(x)}\cos\left[\frac{p_E(x)\xi}{\epsilon}\right], \tag{112}$$

where $p_E(x) = \sqrt{2m(E - V(x))}$ is the classical momentum of the particle, $v_E(x) = p_E(x)/m$ is the classical velocity, and $\theta(E - V(x))$ is a step function that guarantees that $\mathcal{W}_{\mathrm{mc}}(x_1, x_2)$ is nonzero only if the center-of-mass coordinate $x = (x_1 + x_2)/2$ belongs to the allowed region. While this is a stationary state for general potentials, this operator is no longer expected to commute with the Hamiltonian, such that its eigenstates do not exactly correspond to the quantum eigenstates.

Rather than exactly diagonalizing this operator, we can obtain a connection with the WKB approximation by assuming we are far away from the edges of the classically-allowed region, where $E \gg V(x)$ and consider the behaviour for small $\xi = x_1 - x_2$ (the region of $\mathcal{W}$ that is probed by local operators), and approximate

$$p_E(x)\xi = p_E\left(\frac{x_1 + x_2}{2}\right)(x_1 - x_2) \approx S_E(x_1) - S_E(x_2), \qquad S_E(x) = \int_0^x dx' p_E(x'), \tag{113}$$

in which we have introduced the classical action $S_E(x)$, where the lower limit for the integration can be chosen arbitrarily, and take $p_E(x) \approx \sqrt{p_E(x_1)p_E(x_2)}$ to write, in the classically-allowed region and for small $\xi$,

$$\mathcal{W}(x_1, x_2) \approx \frac{2m}{\sqrt{p_E(x_1)p_E(x_2)}}\cos\left[S_E(x_1)/\epsilon - S_E(x_2)/\epsilon\right]/Z \tag{114}$$

$$= \frac{1}{2}\phi_+(x_1)\phi_+(x_2) + \frac{1}{2}\phi_-(x_1)\phi_-(x_2),$$

where we have introduced orthonormalized states

$$\phi_+(x) = \frac{2\sqrt{m}}{\sqrt{Z}}\theta\left[E - V(x)\right]\frac{\cos\left[S_E(x)/\epsilon\right]}{\sqrt{p_E(x)}}, \quad \phi_-(x) = \frac{2\sqrt{m}}{\sqrt{Z}}\theta\left[E - V(x)\right]\frac{\sin\left[S_E(x)/\epsilon\right]}{\sqrt{p_E(x)}}.$$

(115)

As an interesting observation, we note that the Bohr-Sommerfeld quantization of the action,

$$\oint_{E>V(x)} dx\, p_E(x) = \int_{E>V(x)} dx\, \sqrt{2m(E - V(x))} = \left(n + \frac{1}{2}\right)\epsilon\pi,$$

(116)

is here equivalent to demanding that the approximation in (114) does not diverge when $x_1$ and $x_2$ are at opposite edges of the classically-allowed regions where both $p_E(x_1)$ and $p_E(x_2)$ go to zero. The two-dimensional case is introduced in Appendix D. For a more detailed discussion of the connection between the microcanonical distribution and general WKB states we refer the reader to Ref. [46], where the classical and semi-classical limit of Wigner's function is explored for both chaotic and integrable systems.

## C  Explicit derivation for linear and quadratic potentials

### C.1  Linear potential.

Inspired by the Wigner function for the Airy function (see e.g. [46–50]), the explicit eigenstates of the microcanonical ensemble can be found by combining the two identities

$$\int_{-\infty}^{\infty} d\xi\, \mathrm{Ai}(x + \xi/2)\mathrm{Ai}(x - \xi/2)e^{ik\xi} = 2^{2/3}\mathrm{Ai}\left[2^{2/3}(x + k^2)\right],$$

(117)

$$\int_{-\infty}^{\infty} dt\, \mathrm{Ai}(t + x)\mathrm{Ai}(t + y) = \delta(x - y).$$

(118)

The second identity expresses the orthogonality of translated Airy functions, whereas the first was originally obtained in Ref. [47] and was later realized to be part of a larger class of 'projection identities' [48]. Introducing units through $\lambda^3 = \epsilon^2/(2m\alpha)$, these can be rewritten as

$$\int_{-\infty}^{\infty} \frac{d\xi}{\lambda}\mathrm{Ai}\left[\frac{x + \xi/2}{\lambda}\right]\mathrm{Ai}\left[\frac{x - \xi/2}{\lambda}\right]\exp\left[i\frac{p\xi}{\epsilon}\right] = 2^{2/3}\mathrm{Ai}\left[\frac{2^{2/3}}{\lambda}(x + \frac{p^2}{2m\alpha})\right].$$

(119)

The microcanonical distribution for a linear potential can now be written as

$$\begin{aligned}
\delta\left[E - \alpha x - \frac{p^2}{2m}\right] &= \frac{2^{2/3}}{\alpha\lambda}\delta\left[\frac{2^{2/3}}{\alpha\lambda}E - \frac{2^{2/3}}{\lambda}(x + \frac{p^2}{2m\alpha})\right] \\
&= \frac{2^{4/3}}{\alpha^2\lambda^2}\int d\tilde{E}\,\mathrm{Ai}\left[\frac{2^{2/3}}{\alpha\lambda}\left(E - \tilde{E}\right)\right]\mathrm{Ai}\left[\frac{2^{2/3}}{\lambda}\left(x - \frac{\tilde{E}}{\alpha} + \frac{p^2}{2m\alpha}\right)\right] \\
&= \frac{2^{2/3}}{\alpha^2\lambda^3}\int d\tilde{E}\,\mathrm{Ai}\left[\frac{2^{2/3}}{\alpha\lambda}\left(E - \tilde{E}\right)\right] \\
&\qquad\times \int_{-\infty}^{\infty} d\xi\,\mathrm{Ai}\left[\frac{x - \tilde{E}/\alpha + \xi/2}{\lambda}\right]\mathrm{Ai}\left[\frac{x - \tilde{E}/\alpha - \xi/2}{\lambda}\right]\exp\left[i\frac{p\xi}{\epsilon}\right],
\end{aligned}$$

(120)

such that

$$\mathcal{W}(x_1, x_2) = 2\pi\epsilon\frac{2^{2/3}}{\alpha^2\lambda^3}\int d\tilde{E}\,\mathrm{Ai}\left[\frac{2^{2/3}}{\lambda\alpha}(E - \tilde{E})\right]\mathrm{Ai}\left(\frac{x_1 - \tilde{E}/\alpha}{\lambda}\right)\mathrm{Ai}\left(\frac{x_2 - \tilde{E}/\alpha}{\lambda}\right).$$

(121)

## C.2 Harmonic Oscillator.

The eigenvalues and eigenstates of the microcanonical ensemble for the harmonic oscillator can also be analytically obtained, where the eigenstates necessarily correspond to those of the quantum Hamiltonian. These are given by

$$\psi_n(x) = \frac{1}{\left(\pi\xi_0^2\right)^{1/4}} \frac{1}{\sqrt{2^n n!}} H_n(x/\xi_0) \exp\left[-\frac{x^2}{2\xi_0^2}\right], \qquad \xi_0^2 = \frac{\hbar}{m\omega}, \tag{122}$$

and the Wigner function of these states is given by (see e.g. Ref. [51])

$$\begin{aligned} P_n(x,p) &= \int \frac{d\xi}{2\pi\epsilon} \exp\left[i\frac{p\xi}{\epsilon}\right] \psi_n(x+\xi/2)\psi_n(x-\xi/2) \\ &= \frac{(-1)^n}{\epsilon\pi} \exp\left[-\frac{2}{\epsilon\omega}\left(\frac{p^2}{2m} + \frac{1}{2}m\omega^2 x^2\right)\right] L_n\left[\frac{4}{\epsilon\omega}\left(\frac{p^2}{2m} + \frac{1}{2}m\omega^2 x^2\right)\right], \end{aligned} \tag{123}$$

with $L_n$ the Laguerre polynomials. These satisfy the orthonormality relation

$$\delta(x-y) = e^{-(x+y)/2} \sum_{n=0}^{\infty} L_n(x)L_n(y), \tag{124}$$

which can be used to express

$$\begin{aligned} \delta\left[E - \frac{p^2}{2m} - \frac{1}{2}m\omega^2 x^2\right] &= \frac{4}{\epsilon\omega} \exp\left[-\frac{2E}{\epsilon\omega}\right] \exp\left[-\frac{2}{\epsilon\omega}\left(\frac{p^2}{2m} + \frac{1}{2}m\omega^2 x^2\right)\right] \\ &\times \sum_{n=0}^{\infty} L_n\left[\frac{4E}{\epsilon\omega}\right] L_n\left[\frac{4}{\epsilon\omega}\left(\frac{p^2}{2m} + \frac{1}{2}m\omega^2 x^2\right)\right]. \end{aligned} \tag{125}$$

Using the known transform of the oscillator states (123) then leads to

$$\delta\left[E - \frac{p^2}{2m} - \frac{1}{2}m\omega^2 x^2\right] = \sum_{n=0}^{\infty} (-1)^n \frac{4\pi}{\omega} L_n\left[\frac{4E}{\epsilon\omega}\right] \exp\left[-\frac{2E}{\epsilon\omega}\right] P_n(x,p), \tag{126}$$

and we have that for the harmonic oscillator the transform of the microcanonical ensemble can be expanded as (up to a normalization factor $\mathcal{Z} = 2\pi/\omega$)

$$\mathcal{W}_{mc}(x_1, x_2) = \sum_{n=0}^{\infty} (-1)^n \frac{4\pi}{\omega} L_n\left[\frac{4E}{\epsilon\omega}\right] \exp\left[-\frac{2E}{\epsilon\omega}\right] \psi_n(x_1)\psi_n(x_2). \tag{127}$$

The eigenvalues are highly similar to the Wigner function, which can be understood by noting that

$$\begin{aligned} w_n &= \int dx_1 \int dx_2 \mathcal{W}(x_1, x_2) \psi_n^*(x_1)\psi_n(x_2) \\ &= 2\pi\epsilon \int dx \int dp \, \delta\left[E - \frac{p^2}{2m} - \frac{1}{2}m\omega^2 x^2\right] P_n(x,p), \end{aligned} \tag{128}$$

and $P_n(x,p)$ is a function of $\frac{p^2}{2m} + \frac{1}{2}m\omega^2 x^2$, immediately leading to the correct expression for the eigenvalues.

The relation (43) can similarly be obtained by making use of the known transform of the harmonic oscillator states (123), now using the generating function of the Laguerre polynomials [52] as given by

$$\sum_{n=0}^{\infty} t^n L_n(y) = \frac{1}{1-t} e^{-ty/(1-t)}, \qquad |t| < 1. \tag{129}$$

Starting from the expression for the normalized Gibbs ensemble with inverse temperature $\beta$

$$\mathcal{W}(x_1, x_2) = \frac{1}{\mathcal{Z}} \sum_{n=0}^{\infty} e^{-n\beta\epsilon\omega} \psi_n(x_1)\psi_n(x_2), \qquad \mathcal{Z} = \sum_{n=0}^{\infty} e^{-n\beta\epsilon\omega} = \frac{1}{1-e^{-\beta\epsilon\omega}}, \tag{130}$$

its transform is given by

$$P(x, p) = \sum_{n=0}^{\infty} \frac{e^{-n\beta\epsilon\omega}}{\mathcal{Z}} \frac{(-1)^n}{\epsilon\pi} \exp\left[-\frac{2}{\epsilon\omega}\left(\frac{p^2}{2m} + \frac{1}{2}m\omega^2 x^2\right)\right] L_n\left[\frac{4}{\epsilon\omega}\left(\frac{p^2}{2m} + \frac{1}{2}m\omega^2 x^2\right)\right], \tag{131}$$

where the summation can be explicitly evaluated from the generating function (129) by taking $t = -e^{-\beta\epsilon\omega}$ as

$$P(x, p) = \frac{1}{\epsilon\pi\mathcal{Z}} \frac{1}{1 + e^{-\beta\epsilon\omega}} \exp\left[-\frac{2}{\epsilon\omega}\left(\frac{p^2}{2m} + \frac{1}{2}m\omega^2 x^2\right)\right]$$

$$\times \exp\left[\frac{4}{\epsilon\omega} \frac{e^{-\beta\epsilon\omega}}{1 + e^{-\beta\epsilon\omega}}\left(\frac{p^2}{2m} + \frac{1}{2}m\omega^2 x^2\right)\right]$$

$$= \frac{1}{\epsilon\pi\mathcal{Z}} \frac{1}{1 + e^{-\beta\epsilon\omega}} \exp\left[-\frac{p^2}{m\epsilon\omega}\tanh\left(\frac{\beta\epsilon\omega}{2}\right) - \frac{m\omega x^2}{\epsilon}\tanh\left(\frac{\beta\epsilon\omega}{2}\right)\right], \tag{132}$$

returning a Gaussian distribution with

$$\sigma_x^2 = \frac{\epsilon}{2m\omega}\coth\left(\frac{\beta\epsilon\omega}{2}\right), \qquad \sigma_p^2 = \frac{m\epsilon\omega}{2}\coth\left(\frac{\beta\epsilon\omega}{2}\right), \tag{133}$$

or, equivalently,

$$\sigma_x\sigma_p = \frac{\epsilon}{2}\coth\left(\frac{\beta\epsilon\omega}{2}\right), \qquad \frac{\sigma_p}{\sigma_x} = m\omega. \tag{134}$$

The prefactor in $P(x, p)$ can be simplified to read

$$\frac{1}{\epsilon\pi\mathcal{Z}} \frac{1}{1 + e^{-\beta\epsilon\omega}} = \frac{1}{\epsilon\pi} \frac{1 - e^{-\beta\epsilon\omega}}{1 + e^{-\beta\epsilon\omega}} = \frac{1}{\epsilon\pi}\tanh\left(\frac{\beta\epsilon\omega}{2}\right) = \frac{1}{2\pi\sigma_x\sigma_p}, \tag{135}$$

such that the final distribution can be written as a normalized Gaussian distribution with widths set by Eq. (134) as

$$P(x, p) = \frac{1}{2\pi\sigma_x\sigma_p}\exp\left[-\frac{x^2}{2\sigma_x^2} - \frac{p^2}{2\sigma_p^2}\right]. \tag{136}$$

Note that this derivation did not depend on $\beta$ being real, only on $|e^{-\beta\epsilon\omega}| < 1$ in order for the generating function to hold. Now setting $\beta = \tilde{\beta} + \frac{i\pi}{\epsilon\omega}$ with $\tilde{\beta}$ positive, we obtain the expressions from the main text where

$$\sigma_x\sigma_p = \frac{\epsilon}{2}\tanh\left(\frac{\tilde{\beta}\epsilon\omega}{2}\right), \qquad \frac{\sigma_p}{\sigma_x} = m\omega, \tag{137}$$

and

$$\mathcal{W}(x_1, x_2) = \frac{1}{\mathcal{Z}} \sum_{n=0}^{\infty} (-1)^n e^{-n\tilde{\beta}\epsilon\omega} \psi_n(x_1)\psi_n(x_2), \qquad \mathcal{Z} = \frac{1}{1 + e^{-\tilde{\beta}\epsilon\omega}}. \tag{138}$$

# D  Microcanonical ensembles in two-dimensional systems

We can now consider two-dimensional systems, where the WKB approximation no longer holds and the connection with classical stationary distributions is usually made through chaos and non-ergodicity. The 2D microcanonical distribution is given by

$$P(x, p_x, y, p_y) = \frac{1}{4\pi mS} \delta \left( E - \frac{p_x^2}{2m} - \frac{p_y^2}{2m} - V(x, y) \right),$$

$$(139)$$

with $S$ the surface area of the classically-allowed region $E > V(x, y)$. Writing $\vec{r} = (x, y)$ and $\vec{\xi} = (\xi_x, \xi_y)$, the corresponding quasi-density matrix follows as

$$\begin{aligned}
\mathcal{W}(\vec{r} + \vec{\xi}/2, \vec{r} - \vec{\xi}/2) &= \frac{1}{4\pi mS} \int d\vec{p} \, \exp\left[ -i\frac{\vec{p} \cdot \vec{\xi}}{\epsilon} \right] \delta \left( E - \frac{\vec{p}^2}{2m} - V(\vec{r}) \right) \\
&= \frac{1}{4\pi mS} \int_0^{2\pi} d\phi \int_0^\infty dp \, p \, \exp\left[ -i\frac{p|\xi|\cos\phi}{\epsilon} \right] \delta \left( E - \frac{p^2}{2m} - V(\vec{r}) \right) \\
&= \frac{1}{2\pi S} \int d\phi \int dp \, \exp\left[ -i\frac{p|\xi|\cos\phi}{\epsilon} \right] \delta[p - p(\vec{r})]\theta[E - V(\vec{r})],
\end{aligned}$$

$$(140)$$

where we have switched to polar coordinates $\vec{p} = (p\cos\phi, p\sin\phi)$ in the second line and defined $p(\vec{r}) = \sqrt{2m(E - V(\vec{r}))}$. Continuing,

$$\begin{aligned}
\mathcal{W}(\vec{r} + \vec{\xi}/2, \vec{r} - \vec{\xi}/2) &= \frac{1}{2\pi S} \theta[E - V(\vec{r})] \int d\phi \, \exp\left[ -i\frac{p(\vec{r})|\xi|\cos\phi}{\epsilon} \right] \\
&= \frac{1}{S} \theta[E - V(\vec{r})] J_0 \left( \frac{p(\vec{r})|\xi|}{\epsilon} \right),
\end{aligned}$$

$$(141)$$

with $J_0$ a Bessel function of the first kind. Note that this expression was also obtained in M. Berry's original paper [30], introducing what is now known as Berry's conjecture (see also Ref. [46]).

# E  Landau levels

Following Section 5.5, we consider the quasi-density matrix

$$\begin{aligned}
\mathcal{W}(x_1, y_1; x_2, y_2) = \frac{1}{Z} \exp\left[ -\frac{m(x_2 - x_1)^2}{2\epsilon^2\beta} \right] \exp\left[ -\frac{m(y_2 - y_1)^2}{2\epsilon^2\beta} \right] \\
\times \exp\left[ -i\frac{qB}{\epsilon c}(y_2 + y_1)(x_2 - x_1) \right].
\end{aligned}$$

$$(142)$$

To simplify the following derivation, we will neglect the factor $1/Z$. The initial Hamiltonian is translationally invariant along the $x$-direction, such that the quasi-density only explicitly depends on $(x_2 - x_1)$ (and not $x_2 + x_1$) and we can Fourier transform the quasi-density matrix along $x_1$ and $x_2$ to first simplify the problem.

$$\begin{aligned}
\tilde{\mathcal{W}}(k_1, y_1; k_2, y_2) &= \int dx_1 \int dx_2 \, e^{ik_1 x_1} e^{ik_2 x_2} \mathcal{W}(x_1, y_1; x_2, y_2) \\
&= \exp\left[ -\frac{q^2 B^2 \beta}{16mc^2}(y_2 + y_1 - 2\epsilon y_0) \right] \exp\left[ -\frac{m(y_2 - y_1)^2}{2\epsilon^2\beta} \right],
\end{aligned}$$

$$(143)$$

where we defined $y_0 = \epsilon c k_2/(qB)$. This can be further simplified by introducing the cyclotron frequency $\omega = qB/(mc)$. We can now define shifted coordinates $u_i = \sqrt{\frac{m\omega}{\epsilon}}(y_i - y_0)$ to write

$$\tilde{\mathcal{W}}(k_1, y_1; k_2, y_2) = 2\pi\delta(k_1 + k_2)\exp\left[-\frac{\beta\epsilon\omega}{2}\frac{(u_2 + u_1)^2}{4}\right]\exp\left[-\frac{2}{\beta\epsilon\omega}\frac{(u_2 - u_1)^2}{4}\right]. \quad (144)$$

This can be diagonalized by using the following identity from Ref. [53],

$$\frac{1}{\sqrt{\pi(1 - a^2)}}\exp\left[-\frac{1-a}{1+a}\frac{(x+y)^2}{4}\right]\exp\left[-\frac{1+a}{1-a}\frac{(x-y)^2}{4}\right] = \sum_{n=0}^{\infty}a^n\psi_n(x)\psi_n(y), \quad (145)$$

with

$$\psi_n(x) = \frac{1}{\pi^{1/4}}\frac{1}{\sqrt{2^n n!}}H_n(x)e^{-x^2}. \quad (146)$$

Plugging in

$$a = \frac{1 - \beta\epsilon\omega/2}{1 + \beta\epsilon\omega/2} \quad (147)$$

then returns

$$\tilde{\mathcal{W}}(k_1, y_1; k_2, y_2) = 2\pi\delta(k_1 + k_2)\sqrt{\frac{\pi\beta\epsilon\omega}{(1 + \beta\epsilon\omega/2)^2}}\sum_{n=0}^{\infty}\left(\frac{1 - \beta\epsilon\omega/2}{1 + \beta\epsilon\omega/2}\right)^n\psi_n(u_1)\psi_n(u_2). \quad (148)$$

Performing an inverse Fourier transform then returns the expression from the main text (82).

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
