# Peer review of "Quantum eigenstates from classical Gibbs distributions"

_SciPost Physics, doi:SciPost Phys. 10, 014 (2021)_

## Round 2 · Referee Report · Anonymous · 2020-8-28

Strengths

Mini review-styled paper with many details.

Weaknesses

Originality, but compensated by its style, see report.

Report

This paper reviews in principle well known results, but in a style which brings together in a clear language various aspects connected here. Many detailed examples are given, a strengths of this mini review styled paper.

I suggest the authors consider the following minor issues:

(1) ref. 26 spells wrongly the author's name! Its JK Moser! Details of the ref. must be given also.

(2) The introductions links to complementary approaches to the main idea here, both are indeed well known in the literature. hence, I suggest to stress this better, also by citing relevant literature:

- on the first approach (semiclassical expansion a la Weyl-Moyal): Hamiltonian Systems: Chaos and Quantization (Cambridge Monographs on Mathematical Physics) from de Almeida, Alfredo M. Ozorio
- on the second one (truncated Wigner expansion): quantum optics literature by the New Zealand groups, Peter Drummond, H. J. Carmichael, CW Gardinger and others, see e.g. M. J. Werner and P. D. Drummond, J. Comput. Phys. 132, 312 (1997) or C.W. Gardiner, StochasticMethods: A Handbook for the Natural and Social Sciences, Springer Series in Synergetics (Springer-Verlag, Berlin, 2009).

---

## Round 2 · Referee Report · Anonymous · 2020-10-6

Report

Report on 'Quantum eigenstates from classical Gibbs distributions'

The density operator corresponding to the quantum canonical ensemble can be represented by its Wigner function as a real function in the corresponding classical phase space. In the limit of high temperatures (small \beta), this function is accurately approximated by the classical canonical probability density. Thus, the inverse Wigner transform of this classical density can be equated approximately to a spectral decomposition of the projectors onto the eigenstates |n> of the Hamiltonian, weighed by w_n = \exp -\beta E_n. Of course, this is just the spectral decomposition of the evolution operator with imaginary time and temperature replacing Planck's constant. Actually, \hbar is treated as a mobile parameter in semiclassical methods, while here it is replaced by \epsilon. Thus, \epsilon (or \hbar) should also be small for the classical approximation to hold, without the spectral graininess being perceptible. A further step taken here is to transfer, through the inverse Wigner transform, the classical Liouville evolution to the canonical density matrix in equation (19). This is an approximation to the quantum evolution for small \epsilon.

The obvious direct course is to investigate the quantum canonical Wigner function at lower temperatures, but, surprisingly, this paper proceeds in the opposite direction. Not only are the spectral weights obtained for the continuation of the classical density to lower temperatures, but also the eigenstates themselves. No justification is given for the physical relevance of these hybrid classical-quantum constructions, except for the possible implicit assertion that the classical high temperature regime of the Wigner function
provides a competitive method for the evaluation of individual eigenstates. Unfortunately, the presentation is not sufficiently clear to confirm this unequivocally, so as to distinguish an important new metod from a theoretical curiosity. Indeed, the detailed explanation, as to how the stationary wave functions are actually calculated, is missing.

The general equation for the matrix elements of the density matrix in an arbitrary basis of orthogonal eigenstates is given by (50), but there is no special role for Wigner functions in this. Then in section 5.2 the inverse Wigner transform of the classical canonical distribution (71) is treated as a bona fide density matrix, so that its 'exact' eigenvalue equation is (72). Since the distribution is stationary by construction, there is no need to consider explicitly the transform of the Liouville equation. In the rest of this section, it is shown that for sufficiently small \epsilon and \beta, this is approximately equivalent to the stationary Schrödinger equation (with exponential eigenvalues).

Is there any advantage to solve this equation instead of the Schrödinger equation? Are the surprisingly accurate 'classical eigenfunctions', which are compared in the examples,
calculated in this way? One should note that it is fairly standard to compute numerically low lying eigenstates; is there any advantage to compute them in this new way? Could one use this method for high excited states, which are difficult computationally? Even further, could one get at eigenstates of classically chaotic systems in this way?

The recommendation for this paper to be published in a first class journal depends on these issues. Careful consideration should be given to the presentation in a revised version of the paper. The exact results for the linear potential and the harmonic oscillator do not require the solution of a new eigenvalue equation, so they do not prepare the reader for this novel use of the canonical density. It is certainly not true that the Hamilton-Jacobi equation (75) is 'exactly equivalent' to the Schrödinger equation. The formulae for 'General potentials' in section 5.1 are not identical to the ones for the linear potential and the harmonic oscillator in those special cases...

  • validity: -
  • significance: -
  • originality: -
  • clarity: -
  • formatting: -
  • grammar: -

Author:  Pieter W. Claeys  on 2020-12-22  [id 1100]

(in reply to Report 2 on 2020-10-06)

We would like to thank the Referee for their detailed report and reading of the manuscript.

  • The density operator corresponding to the quantum canonical ensemble can be represented by its Wigner function as a real function in the corresponding classical phase space. In the limit of high temperatures (small $\beta$), this function is accurately approximated by the classical canonical probability density. Thus, the inverse Wigner transform of this classical density can be equated approximately to a spectral decomposition of the projectors onto the eigenstates |n> of the Hamiltonian, weighed by $w_n = \exp[-\beta E_n]$. Of course, this is just the spectral decomposition of the evolution operator with imaginary time and temperature replacing Planck's constant. Actually, $\hbar$ is treated as a mobile parameter in semiclassical methods, while here it is replaced by $\epsilon$. Thus, $\epsilon$ (or $\hbar$) should also be small for the classical approximation to hold, without the spectral graininess being perceptible. A further step taken here is to transfer, through the inverse Wigner transform, the classical Liouville evolution to the canonical density matrix in equation (19). This is an approximation to the quantum evolution for small $\epsilon$.

We agree with the Referee. However, the main new results of our manuscript are for the Gibbs distribution (as also made clearer in the revised manuscript), where we now highlight that we are dealing with an expansion in $\beta$ rather than $\epsilon$. In particular, we derived the two leading corrections in $\beta$ to the Gibbs Hamiltonian and explicitly showed how these are of the same order in $\epsilon$ (In other words, this expansion is not an expansion in some dimensionless combination of $\epsilon$ and $\beta$). We also wish to emphasize that in the Gibbs case there is no notion of evolution, i.e. no dynamics. In the quantum case it is clear that the Gibbs distribution corresponds to evolution in imaginary time, but this correspondence is not clear classically. Our result is not given by any quantum action as far as we know, it is different, but in the high temperature limit it indeed becomes equivalent to the quantum partition function, which one can represent through the imaginary time path integral. We stress that the mapping is exact even when $\beta$ is not small and we are not sure which evolution the referee has in mind.

  • The obvious direct course is to investigate the quantum canonical Wigner function at lower temperatures, but, surprisingly, this paper proceeds in the opposite direction. Not only are the spectral weights obtained for the continuation of the classical density to lower temperatures, but also the eigenstates themselves. No justification is given for the physical relevance of these hybrid classical-quantum constructions, except for the possible implicit assertion that the classical high temperature regime of the Wigner function provides a competitive method for the evaluation of individual eigenstates. Unfortunately, the presentation is not sufficiently clear to confirm this unequivocally, so as to distinguish an important new method from a theoretical curiosity. Indeed, the detailed explanation, as to how the stationary wave functions are actually calculated, is missing.

The referee is definitely right: this work is largely driven by theoretical curiosity. Note, however, that the complexity of diagonalizing $\mathcal{W}$ is not larger than the complexity of diagonalizing the appropriate quantum Hamiltonian. We also confirmed this numerically. Within classical mechanics, there are also some questions that are easier to address using the language of quantum mechanics, since wave functions contain the information about the whole probability distribution. For example, entanglement is very difficult to compute classically because of a huge sampling problem, but relatively easy to compute from quantum states. We do not want to make any claims in the paper though, since obviously a lot of work has to be done first. One immediate application is that in the added Section 6 we now show how the usual quantum diagnostics of chaos (level spacing statistics) can also be applied to classical systems. We have now also made clear in the manuscript that all states are calculated using a numerical diagonalization method for a discretized Hamiltonian/$\mathcal{W}$.

  • The general equation for the matrix elements of the density matrix in an arbitrary basis of orthogonal eigenstates is given by (50), but there is no special role for Wigner functions in this. Then in section 5.2 the inverse Wigner transform of the classical canonical distribution (71) is treated as a bona fide density matrix, so that its 'exact' eigenvalue equation is (72). Since the distribution is stationary by construction, there is no need to consider explicitly the transform of the Liouville equation. In the rest of this section, it is shown that for sufficiently small $\epsilon$ and $\beta$, this is approximately equivalent to the stationary Schrödinger equation (with exponential eigenvalues).

We agree and we moved all the discussion of the Liouville equation to the Appendix in order to make the focus of this work more clear. Moreover, we also derived the leading corrections to the Hamiltonian and highlighted how this is not an expansion in $\epsilon$ or some other combination of $\epsilon$ and $\beta$. Moreover, we analyzed the effect of this correction on the tunneling problem, which is not perturbative in $\epsilon$, and found that we improve the accuracy of the tunneling gap by orders of magnitude even when all parameters including $\beta$ are of the order of 1 and the only somewhat small parameter is $\epsilon=0.1$, which is needed simply to have tunneling. With these parameters getting such an accuracy of the tunneling gap is truly remarkable, we think.

  • Is there any advantage to solve this equation instead of the Schrödinger equation? Are the surprisingly accurate 'classical eigenfunctions', which are compared in the examples, calculated in this way? One should note that it is fairly standard to compute numerically low lying eigenstates; is there any advantage to compute them in this new way? Could one use this method for high excited states, which are difficult computationally? Even further, could one get at eigenstates of classically chaotic systems in this way?

As far as we can tell the complexity is identical. We numerically diagonalize matrices of the same size and it takes the same amount of time to diagonalize them by standard routines. Note that we can choose $\epsilon$ at will and use it to our advantage, for example to minimize the number of nonzero components $w_n$. As for chaotic and highly-excited states: the mapping is always exact and we added a new section 6 on chaotic systems, where we diagonalized $\mathcal{W}$ and showed how the BGS and Berry-Tabor conjectures hold in classical setups.

  • The recommendation for this paper to be published in a first class journal depends on these issues. Careful consideration should be given to the presentation in a revised version of the paper. The exact results for the linear potential and the harmonic oscillator do not require the solution of a new eigenvalue equation, so they do not prepare the reader for this novel use of the canonical density. It is certainly not true that the Hamilton-Jacobi equation (75) is 'exactly equivalent' to the Schrödinger equation. The formulae for 'General potentials' in section 5.1 are not identical to the ones for the linear potential and the harmonic oscillator in those special cases...

We hope we addressed the concerns by the referee. We added a lot of additional material (see the first part of the response and the provided list of changes) and significantly changed the presentation of the material in order to improve readability. We believe that the main results of this paper are new and very interesting (at least they seem so to us). At the moment, we of course cannot judge how and where these results will find applications, but we found various surprises in this research going beyond a sheer reproducing of known quantum results to new results in classical systems.

Pieter W. Claeys Anatoli Polkovnikov

---

## Round 2 · Referee Report · Anonymous · 2020-11-2

Strengths

1- A different/fresh perspective on classical quantum correspondence
2- Very pedagogical and intuitive presentation

Weaknesses

Very few,
perhaps that the basic idea in the paper is not original.

Report

The paper elaborates on a beautiful correspondence between classical and quantum mechanics in phase space. Given a classical phase space density P(x,p), one can define a quasi-density matrix W(x1,x2), with respect to a "formal" resolution parameter \epsilon, such that W(x1,x2) corresponds to a proper density matrix when P(x,p) is a Wigner function of a quantum state and \eps is the Planck's constant \hbar. One can now derive quantum formulation of classical mechanics, such as the eigenvectors and the spectrum of the density matrix, writing the corresponding classical Schroedinger equation (which becomes an integral equation, reducing to the usual Schroedinger equation when \eps or \beta are small enough), etc. One can even extend this intriguing correspondence to dynamics and define the Liouvillian propagator for classical quasi-density matrices. Particularly intriguing is the connection between the classical Gibbs phase space distribution and the corresponding quantum eigenfunctions, obtained as eigenvectors of the classical quasi-density matrix. Another intriguing but well discussed issue is the emergence of "negative classical probabilities" for small parameter \eps (smaller than the scale of variation (or uncertainties) of classical probability density).

It is true that most of results discussed here appeared in the literature before, but this paper gives an overarching discussion and a clear physical picture. Even though such a formulation of classical dynamics may not be practical, it can stimulate further interesting studies into quantum classical correspondence, in particular from the viewpoint of quantum chaos. I thus recommend the paper for publication in SciPost Physics.

Requested changes

- The precise mathematical meaning of the spectrum and eigenfunctions of the the classical quasi-density matrix seems unclear. For example, there exist a rigorous formulation of classical mechanics within the Hilbert space, the so-called Koopman-Von Neumann picture. Perhaps the results of the present paper could be phrased or linked to this broad picture. Maybe just a sentence or two would be helpful to give a broad picture.

- How is the classical spectrum related to chaos in two or more dimensions? Is there an extension of Bohigas-Giannoni-Schmidt conjecture to that case? It would be an interesting question for a followup study.

- Top of page 7: typo: particule -> particular

- Text after Eq. (44): The authors discuss that classical probabilities are "oscillatory", in n? Is this true as the formula in the text does not say that, it just says they are negative, if \tilde{\beta}_q is real (as it is said in the text after Eq. (43)).

- Text after Eq. (7): "is" is missing after the first word.

- Caption title of Fig.4: To unformise the style with other figures, add the information that the figure refers to a "quartic potential"

- Caption title of Fig. 7: .. add information that the figure referes to "double well potential".

  • validity: top
  • significance: high
  • originality: good
  • clarity: high
  • formatting: excellent
  • grammar: perfect

Author:  Pieter W. Claeys  on 2020-12-22  [id 1099]

(in reply to Report 3 on 2020-11-02)

We thank the Referee for their nice and constructive comments. We, however, would disagree with the idea that most results appeared before. It is true that, while we were working on the paper, we found out that the formalism of mapping $P(x,p)$ to $\mathcal{W}(x_1,x_2)$ and the associated Schrödinger equation appeared in earlier literature. However, we decided to include this material because those works are surprisingly unknown and several important concepts such as the details of the representation of phase space variables through Hermitian operators were missing or only mentioned briefly in the literature (as far as we are aware). This part of the paper was indeed intended to be a comprehensive introduction to earlier results, with some relatively small new results. We agree that the connection between the classical Gibbs phase space distribution and the quantum eigenfunctions is particularly intriguing, and this was and is intended to be the focus of the manuscript — where we believe most results to be new. With this in mind, we have reworked the presentation of the manuscript and included additional results on the Gibbs ensemble.

As for the requested changes: - We added a comment on the Koopman-von Neumann picture. Although similar in spirit, we did not find a direct relevance of this formulation to our work. - We thank the referee for the suggestion on chaos and the Bohigas-Giannoni-Schmidt conjecture, and we added a new Section 6 to address this question. This conjecture (as well as the Berry-Tabor conjecture) appears to work beautifully if we analyze the spectrum of $\mathcal{W}$. This observation highlights that these conjectures are intrinsic properties of the spectrum of the $\mathcal{W}$-matrix, which are not related to the approximate mapping to the quantum Hamiltonian. Alternatively (although we decided not to include this), if we take the classical eigenstates and compute the energy spectrum as the expectation value of the Hamiltonian, the level repulsion and the expected level spacing statistics disappear even at small $\beta$, since small differences between classical and quantum eigenstates are sufficient to destroy the level repulsion in this measure. - This typo has been corrected. - There was a factor $(-1)^n$ missing in the inline equation (other equations were correct). This has been corrected. - This has been corrected. - This has been corrected. - This has been corrected.

Pieter W. Claeys Anatoli Polkovnikov

---

## Round 3 · Referee Report · Anonymous (Referee 3) · 2021-1-5

Report

I believe that the authors have adequately taken care of all the points raised, hence I recommend the manuscript for publication.

---

## Round 3 · Referee Report · Anonymous (Referee 2) · 2021-1-7

Strengths

Very original work, which increases our understanding of the complex relation between classical and quantum mechanics.

Report

The authors have substantially clarified the text and provided extentions that are of great value. Publication of this new veresion is fully recommended.

The single fault that seems to have slipped through is that equation (53) is not "exactly equivalent to the Schrödinger equation".

Another improvement could be to substitute the notation for the normalization integral Z_x in (14), since it tends to mislead the reader into attributing an
x-dependence to it.

---

## Round 3 · Author Response

We thank the referees for providing valuable feedback. Following their suggestions we rearranged the material to highlight the new results related to the classical Gibbs ensemble. We rearranged the material such that original derivations and the main results of the paper appear sooner, moving some of the introductory material and the results on microcanonical ensembles to the Appendices. Next to some minor textual changes, we also added some additional topics suggested by the referees and also by our other colleagues, in particular by Sir. Michael Berry. The key changes can be found below.

---

## Round 3 · List of Changes

• We have rearranged the structure of the paper such that the new results on the Gibbs distribution appear earlier. Introductory sections on the dynamics and microcanonical ensemble have been moved to Appendix.

  • We have added a derivation of the two leading-order corrections in $\beta$ to the Gibbs Hamiltonian (Sec. 4.2). These corrections highlight that this is {\emph not} a semiclassical expansion: different orders in $\beta$ have the same order in $\epsilon$. We tested these corrections in the example of tunneling and even for $\beta=1$, i.e. when the small parameter is not that small, we observed that the relative mistake in the tunneling gap can be reduced by orders of magnitude by taking into account higher-order corrections (see the inset in Fig. 6).

  • We have added a derivation of the Gibbs Hamiltonian for a particle in the presence of a vector potential and showed how, to leading order in $\beta$, the resulting Gibbs Hamiltonian again agrees with the quantum Hamiltonian. For this reason, all Berry-type phases encoded in the quantum states also appear to be contained in the classical eigenstates, extending the applicability of this formalism. We illustrated the agreement between quantum and classical eigenstates for a confined particle particle in a 2D potential in the presence of a magnetic field and found excellent correspondence both in terms of absolute values and the phases of the wave functions (Fig. 9). Furthermore, we also added an exact analytic derivation of Landau Levels from the classical Gibbs ensemble in the presence of a constant magnetic field (Sec. 5.5.).

  • We have included a discussion of chaotic and integrable two-dimensional potentials, which was independently suggested by several people including the third referee. We found that the classical spectrum indeed satisfies the BGS and Berry-Tabor conjectures (Sec. 6). It is remarkable that having the Gibbs distribution alone in hand and without analyzing any dynamics, one can determine whether the classical system is chaotic or not.

  • Minor textual changes following the Referees' comments (including corrected typos, an explanation of how all numerical results were obtained, and a short comment on the Koopman-von Neumann construction).

---

## Editorial Decision

published